# Effect of volcanic emissions on clouds during the 2008 and 2018 Kilauea degassing events

Katherine H. Breen[1, 2], Donifan Barahona[1], Tianle Yuan[1, 3], Huisheng Bian[1, 3], and Scott C. James[4]

[1]NASA, Goddard Space Flight Center, Greenbelt, MD, USA
[2]Universities Space Research Association, Columbia, MD, USA
[3]Joint Center for Earth Systems Technology. University of Maryland, Baltimore County, MD, USA
[4]Baylor University, Departments of Geosciences and Mechanical Engineering, Waco, TX, USA

**Correspondence:** Donifan Barahona (donifan.o.barahona@nasa.gov)

**Abstract.**

Volcanic eruptions in otherwise clean environments are "natural experiments" where the effects of aerosol emissions on clouds and climate can be partitioned from meteorological variability and anthropogenic activities. In this work, we combined satellite retrievals, reanalysis products, and atmospheric modeling to analyze the mechanisms of aerosol-cloud interactions during two degassing events at the Kilauea Volcano in 2008 and 2018. The eruptive nature of the 2008 and 2018 degassing events was distinct from long-term volcanic activity for Kilauea. Although previous studies assessed the modulation of cloud properties from the 2008 event, this is the first time such an analysis has been reported for the 2018 event and that multiple degassing events have been analyzed and compared at this location. Both events resulted in significant changes in cloud effective radius and cloud droplet number concentration that were decoupled from local meteorology, and in line with an enhanced cloud albedo. However it is likely that the effects of volcanic emissions on liquid water path and cloud fraction were largely offset by meteorological variability. Comparison of cloud anomalies between the two events suggested a threshold response of aerosol-cloud interactions to overcome meteorological effects, largely controlled by aerosol loading. In both events, the ingestion of aerosols within convective parcels enhanced the detrainment of condensate in the upper troposphere resulting in deeper clouds than observed under pristine conditions. Accounting for ice nucleation on ash particles led to enhanced ice crystal concentrations at cirrus levels and a slight decrease in ice water content, improving the correlation of the model results with the satellite retrievals. Overall, aerosol loading, plume characteristics, and meteorology contributed to changes in cloud properties during the Kilauea degassing events.

## 1 Introduction

Aerosol emissions influence Earth's climate both directly and indirectly. The direct effect involves scattering and absorption of thermal and solar radiation by atmospheric aerosols, while indirect effects involve alteration of the microphysical properties and the global distribution of clouds (Boucher et al., 2013; Twomey, 1977). Both liquid and ice clouds are susceptible to aerosol emissions that can alter their microphysical (i.e., particle size distribution and albedo) and macrophysical properties (liquid and ice water content, cloud lifetime, and cloud fraction) (Lohmann and Feichter, 2005; Boucher et al., 2013; Seinfeld et al., 2016).

These effects, collectively known as aerosol indirect effects (AIEs), may offset a significant fraction of the warming induced by
greenhouse gas emissions, yet their magnitudes are poorly constrained (Boucher et al., 2013). Additionally, cloud formation is
a complex and nuanced physical process occurring on scales far smaller than those resolved by climate models, and the precise
feedback mechanisms influencing AIEs across various timescales are not fully understood (Boucher et al., 2013; Klein et al.,
2013; Malavelle et al., 2017; Yuan et al., 2011).

The presence of cloud condensation nuclei, (CCN; typically sulfates, organics, and nitrate particles) in the atmosphere
indirectly impacts Earth's net radiative balance by increasing the number of cloud droplets, hence altering the scattering and
absorption of incoming solar radiation. This effect, historically referred as the first AIE, or "Twomey" effect (Twomey, 1977),
represents a change in Earth's albedo and results in net radiative cooling. Smaller droplets are also less efficient at coalescing
into rain-bearing clouds. Non-precipitating clouds have a longer lifetime and thereby provide extended coverage, which is
known as the second AIE (Albrecht, 1989). Aerosols can also act as ice nucleation particles (INPs; typically dust, soot, and
organics), modifying cloud properties at low temperature (Lohmann and Feichter, 2005). Besides these effects, CCN and INP
ascend within convective parcels and determine, to a large extent, the onset of precipitation, modifying the release of latent
heat within convective clouds (Koren et al., 2005).

Satellite datasets facilitate global observational monitoring of meteorology, ambient aerosol concentrations and distributions, and cloud properties. However, inferring aerosol–cloud interactions (ACIs) from satellite retrievals is difficult due to the
concurrent influence of meteorological effects (i.e., variation in temperature, water vapor, and winds from large-scale forcing)
on clouds (Lohmann and Feichter, 2005; Gryspeerdt et al., 2017). From a modeling perspective, the relevant scale for ACIs
(i.e., tens to hundreds of meters) is typically unresolved in atmospheric general circulation models (AGCMs; Boucher et al.,
2013), where grid resolution is coarser (generally thousands of meters). The parameterization of ACIs in AGCMs is largely
dependent on theory with many assumptions involved, and model outputs are difficult to validate directly against satellite
retrievals due to differences in resolution.

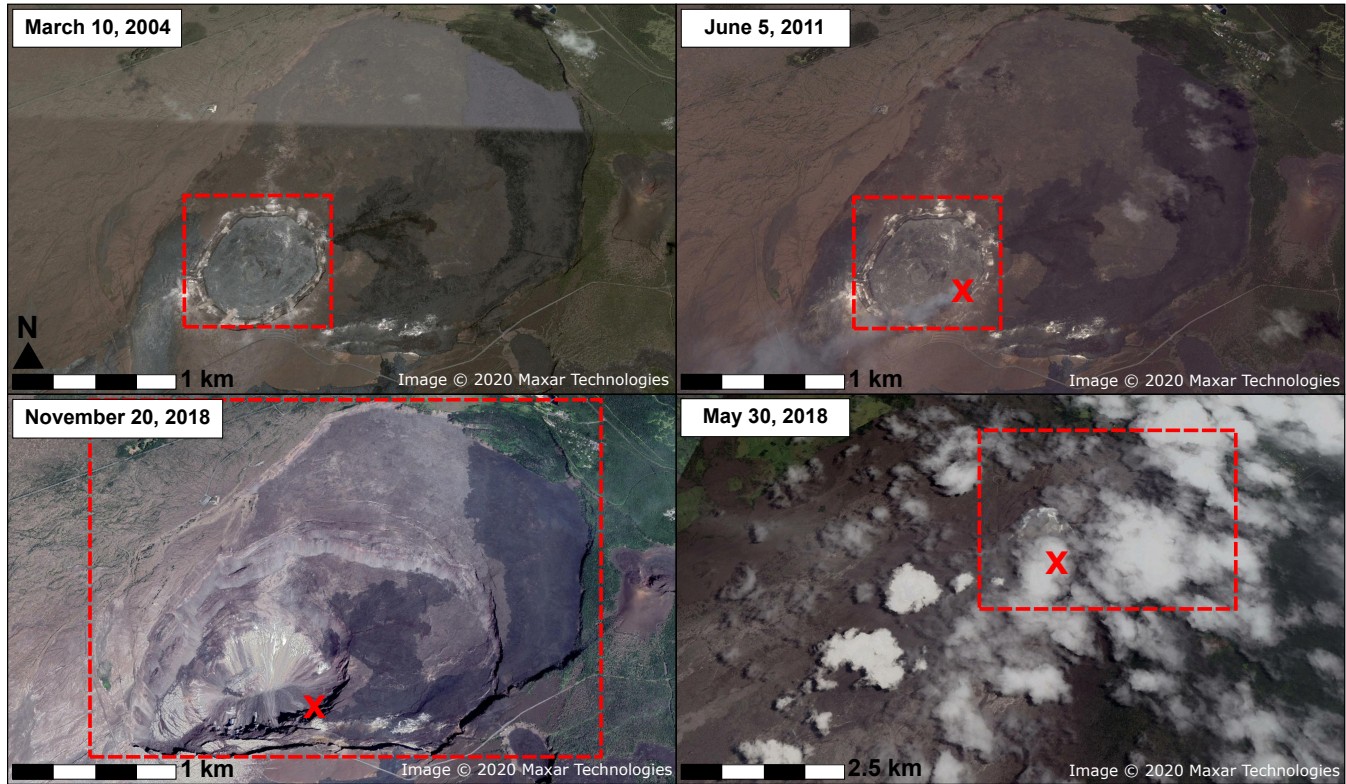

**Figure 1.** © Google Earth imagery of the Kilauea summit crater before and after each degassing event. Clockwise from top left: pre-2008 degassing; post-2008 degassing, during 2018 peak degassing, and post-2018 degassing. The image from June 5, 2011 shows the summit vent, which was not present prior to the 2008 event, passively degassing during an eruptive lull. The image from May 30, 2018 was taken on a cloudy day during the 2018 event, several days after peak observed $SO_2$ emissions ($>100\,\mathrm{kt\,day^{-1}}$) and plume height ($<8\,\mathrm{km}$) (Kern et al., 2020). The red boxes approximately outline the summit crater while the "X"s indicate the degassing vent.

Volcanic degassing events in otherwise "clean" environments, where anthropogenic aerosol emissions are minimal, have been used as "natural" experiments to identify characteristics of ACIs in satellite retrievals and to evaluate AIEs in GCMs. This is well-exemplified at the Kilauea volcano (Fig. 1) in the Hawaiian Islands (Eguchi et al., 2011; Yuan et al., 2011; Beirle et al., 2014; Malavelle et al., 2017). Not only are local anthropogenic emissions low in the local environment surrounding

Kilauea, but because the volcano is situated on an island approximately 1000 miles from the nearest major anthropogenic emissions source (U.S. West coast), there is a low likelihood of multiple emission sources of similar magnitude, therefore the climatic effects of volcanic emissions may be evaluated against relatively pristine background conditions. Kilauea has had long-term monitoring of emissions, seismic activity, and eruptive behavior over the historical record. Being a low-altitude volcano, Kilauea also provides a unique opportunity to study ACIs for liquid and ice cloud phases, since the injection height

of the emissions is mainly controlled by the strength of the eruption.

In this work, we assessed the effects of $SO_2$ and ash aerosols emissions on liquid and ice cloud formation during two volcanic degassing events from Kilauea in June 2008 and May 2018 (Figure 1). In both of these events, atmospheric aerosol loading increased by several orders of magnitude over a few days and remained high for at least two to three months. The volcanic and

seismic activity during peak emissions in 2008 and 2018 has been well-characterized (Nadeau et al., 2015; Neal et al., 2019; Elias and Sutton, 2012; Elias et al., 2020; Wilson et al., 2008; Orr and Patrick, 2009; Orr et al., 2013; Patrick et al., 2019; Kern et al., 2020), revealing that emission rates of $SO_2$ during the 2018 event were conservatively, at least $\geq 5\times$ higher than 2008 peak degassing emissions (Kern et al., 2020; Elias et al., 2020). Analysis of the satellite record has shown significant anomalies in aerosol loading and liquid cloud properties during the 2008 event resulting from volcanic emissions (Eguchi et al., 2011; Yuan et al., 2011; Beirle et al., 2014), in agreement with modeling studies (Malavelle et al., 2017). Tang et al. (2020) showed significant effects of the 2018 Kilauea emissions on air quality. However analyses of ice clouds and of the impacts of the 2018 event on cloud properties and evolution has not yet been reported. This work constitutes a substantial contribution towards identifying ACI signatures for liquid and ice macro and microphysical processes and their sensitivity to aerosol loadings and CCN/INPs.

This work is organized as follows. Section 1.1 provides an overview of each degassing event. Satellite datasets used in the analysis are described in Section 2, and the modeling approach is described in Section 3. Section 4 discusses the results of our analysis with emphasis on liquid and ice cloud interactions. Finally, concluding remarks are given in Section 5.

## 1.1 Kilauea Degassing Events

Kilauea is an active basaltic shield volcano located on the Island of Hawai'i. Volcanic plumes on Kilauea are generally low-altitude ($< 10\,\mathrm{km}$) because the gentle slopes of the volcano provide little protection from strong trade winds and because of the shallow depth of the summit crater ($< 100\,\mathrm{m}$) (Elias et al., 2018). Since March 2008, volcanic activity on Kilauea has been marked by summit degassing and flank eruptions with increasing intensity and frequency (Nadeau et al., 2015). Degassing events were likely triggered by rockfalls related to vent widening and/or seismic activity, which then disturbed the lava lake surface, beneath which a layer of gas had accumulated (Orr and Patrick, 2009; Orr et al., 2013). Degassing events produced variable volumes of tephra (ash), and some larger explosive events scattered lithic material (rock and ash) as much as 50 ha surrounding the vent and produced an ash-rich plume (Elias and Sutton, 2012; Elias et al., 2020; Nadeau et al., 2015). The summit of Kilauea has been in an eruptive state since 2008, and the degassing events of 2008 and 2018 represent brief periods of increased volcanic activity and $SO_2$ emissions resulting in an optically denser plume relative to passive degassing. Some studies have noted residual effects of the plume downwind of Kilauea long after violent eruptions have ceased (Businger et al., 2015; Pattantyus et al., 2018).

From March to August 2008, summit activity was characterized by degassing bursts from lava in the enlarging vent cavity and small explosive events. During degassing events, the estimated $SO_2$ plume height ranged from 1,200 to 2,500 m above sea level and $SO_2$ emissions exceeded 10,000 tonnes per day (Elias et al., 2020). Observational and modeling studies revealed a significant departure of cloud droplet size and optical thickness from their climatological values during the 2008 event, consistent with increased concentrations of atmospheric aerosols within the $SO_2$ plume from the Kilauea summit crater (Eguchi et al., 2011; Yuan et al., 2011; Beirle et al., 2014; Mace and Abernathy, 2016; Malavelle et al., 2017). Figure 1 (top row) shows images before and after the 2008 degassing event. After the 2008 event, the vent can be seen in the SE corner of the summit crater passively degassing from the lava lake during an eruptive pause.

From May to July 2018, summit eruptions and effusive activity in the eastern rift zone (ERZ) were common. Summit activity ejected lithic material and ash $\approx 2,000$ to $8,100$ m above the summit vent and $SO_2$ concentrations were elevated (Neal et al., 2019; Kern et al., 2020). ERZ eruptive events (lava fountains $< 80$ m) were associated with $SO_2$ emissions of $\geq 100,000$ tonnes per day (Neal et al., 2019; Kern et al., 2020; Elias et al., 2018). Multiple cycles of ongoing short pulses of eruptive degassing activity at the vent and longer periods of effusive volcanism in the ERZ caused ongoing aerosol emissions (Patrick et al., 2019). While in 2008, most degassing occurred at the summit vent, in 2018 degassing largely occurred in the ERZ and drained lava beneath the summit vent causing massive deformation of the summit crater as shown in Figure 1, lower left. Additionally, the ocean entry of 2018 ERZ eruptions caused large clouds of vaporized HCl and water vapor to ascend with the plume (Kern et al., 2020). Overall, the height and geologic composition of the plume in 2018 were similar to more violent, siliclastic eruption types (i.e., Mt. St. Helens; Mastin et al., 2009), therefore the 2018 event represented a distinct departure from long-term (decadal) trends. Figure 1 (bottom row) shows the summit crater before, during, and after peak eruptive activity in 2018 and indicates optically denser clouds within the aerosol plume with respect to surrounding clouds during peak degassing (Fig. 1, lower right).

## 2  Data

Data from NASA's MODIS instrument and from the Cloud-Aerosol Lidar and Infrared Pathfinder Satellite Observation (CALIPSO) retrievals were used to assess both horizontal and vertical cloud modification, respectively, as done in previous studies for liquid clouds (Yuan et al., 2011; Malavelle et al., 2017; Eguchi et al., 2011; Beirle et al., 2014; Mace and Abernathy, 2016). The MODIS Aqua Aerosol Cloud Monthly collection 6 L3 Global $1°$ datasets (MYD08_M3) were acquired from the Level-1 and Atmosphere Archive & Distribution System (LAADS) Distributed Active Archive Center (DAAC) (https://ladsweb.nascom.nasa.gov/). Our analysis used MODIS effective radius ($R_{eff}$; liquid, ice), cloud optical depth (COD; liquid, ice), cloud water path (liquid (LWP), ice (IWP), and total (TWP)), aerosol optical depth (AOD), and cloud fraction (CF). All variables are defined in Table 1. The MODIS cloud droplet number concentration (CDNC) climatology (2003–2015) was obtained from Bennartz and Rausch (2017). For 2018, CDNC were calculated using the method of Bennartz (2007). Gridbox mean LWP and IWP were calculated by scaling the MODIS product using the retrieved liquid and ice cloud fractions, respectively. No significant differences were found by scaling using either daily data (MYD08_D3) aggregated into monthly means (Malavelle et al., 2017) or monthly products (MYD08_M3), therefore monthly MODIS products were used in the analysis. MODIS anomalies were calculated as the three-month average during peak degassing minus the long-term mean (2003–2015, excepting 2008). Missing values in MODIS data, primarily found in CDNC and ice products, were smoothed using cubic spline nearest-neighbor interpolation and a Gaussian filter. Uncertainty of MODIS retrievals is discussed in Hubanks et al. (2015) and Bennartz (2007); Bennartz and Rausch (2017) (latter is for CDNC only).

Vertical cloud fraction profiles from the GCM Oriented Cloud CALIPSO Product (CALIPSO-GOCCP) were obtained for comparison with model results. The CALIPSO-GOCCP dataset is developed with CALIPSO L1 data at full horizontal resolution (330 m) and vertical resolution typical for most GCMs (40 levels; $\Delta z = 480$ m) (Chepfer et al., 2010). Instantaneous

**Table 1.** Variable definitions and acronyms.

| Variable | Definition | Units |
|---|---|---|
| ACRI_SNOW | Ice crystal accretion by snow tendency | $cm^{-3}\ s^{-1}$ |
| ACRL_(RAIN,SNOW) | Accretion of liquid by rain/snow tendency | $cm^{-3}\ s^{-1}$ |
| AOD | Aerosol optical depth | – |
| AUT | Liquid autoconversion tendency | $cm^{-3}\ s^{-1}$ |
| AUTICE | Ice autoconversion tendency | $cm^{-3}\ s^{-1}$ |
| CCN | Cloud condensation nuclei | $m^{-3}$ |
| CDNC | Cloud droplet number concentration | $m^{-3}$ |
| CF | Cloud fraction | – |
| CNV_DQLDT | Total detrained condensate tendency | $mg\ kg^{-1}\ s^{-1}$ |
| COD | Cloud optical depth | – |
| COND | Liquid condensation tendency | $kg^{-1}kg^{-1}$ |
| DEP | Ice crystal growth tendency | $cm^{-3}s^{-1}$ |
| DCNVI | Ice convective detrainment tendency | $cm^{-3}\ s^{-1}$ |
| DCNVL | Liquid convective detrainment tendency | $cm^{-3}\ s^{-1}$ |
| EVAP | Droplet evaporation tendency | $cm^{-3}\ s^{-1}$ |
| HM | Ice splintering tendency | $cm^{-3}\ s^{-1}$ |
| ICENUC | Ice nucleation tendency | $cm^{-3}\ s^{-1}$ |
| ICNC | Ice crystal number concentration | $L^{-1}$ |
| INP | Ice nucleating particles | $m^{-3}$ |
| IWP | Ice water path | $g\ m^2$ |
| LWP | Liquid water path | $g\ m^2$ |
| MELT | Ice/snow melt tendency | $kg^{-1}kg^{-1}$ |
| NHET_IMM | Ice nucleation by immersion freezing | $cm^{-3}$ |
| SDM | Ice sedimentation tendency | $kg^{-1}kg^{-1}s^{-1}$ |
| SCF | Supercooled cloud fraction | – |
| $R_{\mathrm{eff}}$ | Effective radius | μm |
| TWP | Total water path | $g\ m^2$ |
| $Q_{\mathrm{liq}}$ | Liquid mixing ratio | $mg\ kg^{-1}$ |
| $Q_{\mathrm{ice}}$ | Ice mixing ratio | $mg\ kg^{-1}$ |
| WBF | Weneger-Bergeron-Findeisen process (ice) | $kg^{-1}kg^{-1}$ |
| WBFSNOW | Weneger-Bergeron-Findeisen process (snow) | $kg^{-1}kg^{-1}$ |

profiles of the lidar scattering ratio are computed and used to infer the vertical and horizontal distributions of cloud fraction. The seasonal mean uncertainty (2006–2008) for GOCCP cloud fraction is $\leq 0.05$ above the boundary layer (Chepfer et al., 2010). The dataset may be used for comparison either directly to AGCM output or to AGCM and LiDAR-simulator output, and is comparable to other standard cloud fraction climatologies (Chepfer et al., 2010). Comparisons between MODIS collection 6 and CALIPSO show agreement between MODIS cloud-phase partitioning and CALIPSO column-wise cloud profiles

(Marchant et al., 2016). GOCCP was used primarily to assess anomalies on the vertical structure of cloud fraction and phase partitioning during both events.

Volcanic $SO_2$ emissions were constrained by observations from the Ozone Monitoring Instrument (OMI) on-board NASA's EOS/Aura spacecraft (Carn et al., 2015). For Kilauea, this dataset only provides "constant" annual $SO_2$ emission rates. For

the 2008 event, we replaced this data set with daily varying emissions (Carn et al., 2017; Yang, 2017). Daily emissions for 2018 were obtained from Li et al. (2020). Missing values were replaced with Ozone Mapping and Profiling Suite data (https://so2.gsfc.nasa.gov/) whenever possible, otherwise the nearest real data point was used. $SO_2$ ( $cm^{-2}$) vertical column density data were converted to emission rates (kg $SO_2$ $s^{-1}$) following the approach of Beirle et al. (2014, Cf. Fig. 6). For uncertainty quantification of OMI retrievals, see Carn et al. (2016) and Carn et al. (2017).

## 3 Methods

To help explain the observed changes in cloud properties during the 2008 and 2018 Kilauea degassing events, and to what these changes were sensitive, we undertook a set of AGCM numerical experiments (Table 2) constrained by observed $SO_2$ emissions (Carn et al., 2015) and by MERRA-2 (Gelaro et al., 2017). Our objective was to generate a close representation of the clouds formed during each event to understand how aerosol emissions impacted cloud microphysics and evolution.

### 3.1 GEOS Model Description

The NASA's Global Earth Observing System (GEOS), version 5, was used to analyze and understand the observed modifications to cloud properties during the 2008 and 2018 Kilauea events (Barahona et al., 2014; Molod et al., 2015). GEOS consists of a set of components that numerically represent different aspects of the Earth system (atmosphere, ocean, land, sea-ice, and chemistry), coupled following the Earth System Modeling Framework (https://gmao.gsfc.nasa.gov/GEOS_systems/). For this work, the AGCM configuration of GEOS was used. Atmospheric transport of water vapor, condensate and other tracers, and associated land-atmosphere exchanges were computed explicitly, whereas sea-ice and sea surface temperature (SST) were prescribed as time-dependent boundary conditions (Reynolds et al., 2002; Rienecker et al., 2008).

Transport of aerosols and gaseous tracers such as CO were simulated using the Goddard Chemistry Aerosol and Radiation model (GOCART) (Colarco et al., 2010), which interactively calculates the transport and evolution of dust, black carbon, organic material, sea salt, and $SO_2$. Dust and sea salt emissions are prognostic whereas biomass burning and antropogenic emissions of $SO_2$, black carbon, and organic carbon are obtained from the Modern Era Retrospective Reanalysis for Research and Applications-Version 2 (MERRA-2) dataset (Randles et al., 2017). GOCART explicitly calculates the chemical conversion of sulfate precursors (dimethylsulfide, or DMS, and $SO_2$) to sulfate. The aging of carbonaceous aerosol is represented by the conversion of hydrophobic to hydrophilic aerosols using an e-folding time of 2 days (Chin et al., 2009). Using the evolving meteorological fields from GEOS, for each time step GOCART simulates the advection (using a flux-form semi-Lagrangian method, (Lin and Rood, 1996)), convective transport, and the wet and dry deposition of aerosol tracers. The calculation of AOD is a function of aerosol size distribution, refractive indices, and hygroscopic growth. Each aerosol type is assumed to be externally mixed. Size distributions are prescribed for different types using 5 bins for dust and sea salt, and single lognormal modes for other aerosol components (Colarco et al., 2010; Chin et al., 2009). This approach was also employed to estimate the aerosol number concentration used in the calculation of aerosol-cloud interactions (Barahona et al., 2014).

Cloud microphysics in GEOS is described using a two-moment scheme where the mixing ratio and number concentration of cloud droplets and ice crystals are prognostic variables (Barahona et al., 2014; Morrison and Gettelman, 2008). The two-moment microphysical model links aerosol emissions to cloud properties and predicts the mixing ratio, number concentration, and effective radius of cloud liquid and ice, rain, and snow for stratiform clouds i.e., cirrus, stratocumulus (Morrison and Gettelman, 2008), and convective clouds (Barahona et al., 2014). Cloud droplet activation is parameterized using the approach of Abdul-Razzak and Ghan (2000). Ice crystal nucleation is described using a physically based analytical approach (Barahona and Nenes, 2009a) that included homogeneous and heterogeneous ice nucleation and their competition. Heterogeneous ice nucleation in the immersion and deposition modes follows Ullrich et al. (2017). Vertical velocity fluctuations were constrained by non-hydrostatic, high-resolution global simulations (Barahona et al., 2017). GEOS has been shown to reproduce the global distribution of clouds, radiation, and precipitation in agreement with satellite retrievals and *in situ* observations (Barahona et al., 2014).

### 3.2 Description of Simulation Experiments

We performed several global integrations of GEOS to best isolate the effects of volcanic aerosol emissions on cloud development. Each simulation was initialized on January 1$^{st}$ of each year and run at a nominal horizontal resolution of 0.5°with 72 vertical levels. The time step was set to $450\,s$ to resolve the large-scale transport of aerosol and condensate. Cloud microphysics was sub-cycled twice each time step to account for unresolved, fast microphysical processes such as CCN activation (Morrison and Gettelman, 2008). For each event, control runs (identified as 1×, Table 2) used the default model and emissions as described in Section 2. A second set of simulations was performed for each event removing the emissions from Kilauea (0×) to represent background conditions unaffected by the degassing events. Sensitivity studies were also performed. A five-fold actual emissions run (2008_5×) was performed to compare the 2008 and 2018 events (the 2018 event emitted $\geq 5$ times the aerosol load of the 2008 event) to assess the similarities between the events with respect to increased aerosol loading. Simulations to test the effect of SO$_2$ injection height (i. e. distribution of emissions constrained by plume height) on cloud microphysics were also performed (2008_PH2km and 2018_PH4km). Finally, another numerical experiment was carried out for the year 2018 (2018_PH4km_ash), where the effects of ash as active INP were investigated, as described in Appendix A. Besides these experiments, long-term model integration (2000–2017) was performed to represent the climatology of the model (GEOS$_{CLIM}$). The simulation experiments are summarized in Table 2.

To account for model drift all simulations were run in "replay" mode, where pre-computed analysis increments from MERRA-2 were applied to nudge the model state (i.e., horizontal winds and temperature) to the reanalysis every six hours. The replay technique is more stable and has lower numerical drift than regular nudging (Takacs et al., 2018). Because the two-moment cloud scheme used in this work differed from the single-moment scheme used in MERRA-2, water vapor was not replayed and instead left to evolve with the model physics. Aerosol concentrations are indirectly constrained by the reanalysis since their transport and evolution, as well as the emission of dust and sea salt, depend on the model state (i.e., winds and temperature). The emission of sulfate precursors (SO2) is constrained using satellite retrievals as described in Section 3.1. However, aerosol concentrations were not directly nudged to the MERRA-2 product, even though the aerosol increments are

also available (Randles et al., 2017). Doing so would have limited the response of clouds to aerosol (via aerosol activation) and
vice-versa, the response of aerosols to cloud formation and precipitation (via scavenging). Running in replay mode ensured
that the effects of meteorological variability were reduced and that our simulations reproduced the assimilated atmospheric
state as closely as possible. Replaying temperature ($T$) also minimized the role of direct and semi-direct effects in modifying
cloud properties. Hence, our analysis focuses on the evolution of cloud microphysical properties.

The altitude of the Kilauea summit crater is $\approx 1200\,\text{m}$, while the top of the well-mixed boundary layer is around 2 km. For
this reason, we allow the degassing volcano emissions to be distributed within 1 km above the summit crater and constrained
by the boundary layer for all 1× and 0× experiments. For alternate experiments, we set an explicit volcanic aerosol injection
height approximating the mean maximum plume height observed during each event (2 km in 2008 (Elias and Sutton, 2012;
Elias et al., 2020; Eguchi et al., 2011)), 4 km in 2018 (Neal et al., 2019)). In this work, we assumed that the source elevation
of emissions (ERZ vs. summit) was irrelevant during peak emissions, but that the injection height of aerosols directly into the
troposphere at different altitudes (below or above boundary layer processes) influenced cloud microphysics and macrophysical
characteristics for liquid and ice clouds. We also assumed the primary aerosol to be $SO_2$ with some percentage of ash, although
it is likely that sea salt contributed to ACIs in liquid clouds and this is recommended for inclusion in future parameterizations.

### 3.3 Analysis Method

MODIS retrievals and the 1× GEOS experiments were compared against the long-term climatology excluding 2008 during
peak degassing periods for each event. The 1× simulations were also compared against the 0× results for each year (1×−0×).
In this way, we assessed the effects of volcanic aerosols on cloud formation separated from natural meteorological variability.
This also allowed for the assessment of whether passive degassing effects present in GEOS$_{\text{CLIM}}$ were significant contributors
to ACIs as opposed to active degassing events in 2008 and 2018. The GEOS$_{\text{CLIM}}$ run captured meteorological variability,
while the 0× scenario represents an alternative reality without volcanic emissions and with the same meteorological state as
the 1× runs. Correlation between observations and the 1×−0× anomaly would indicate that volcanic emissions forced ACIs
that were decoupled from meteorology (i. e. the observed effects would not have been present without elevated emissions).
Conversely, correlation with the GEOS$_{\text{CLIM}}$ anomalies would indicate that the observed anomaly cannot be partitioned from
meteorological variability in the regional climate.

Anomalies for each degassing event were calculated as the seasonal mean during peak eruptive periods with long-term
seasonal averages removed. We focused on the boreal summer (June-July-August, JJA) for the 2008 degassing event and the
transition from the boreal spring to summer (May-June-July, MJJ) for 2018 when emissions from both events were highest and
active eruptive events projecting ash and lithics into the volcanic plume occurred. To assess anomalies at the source relative to
areas outside of the plume, normalized zonal mean anomalies were calculated as

$$v_{\text{lat,norm}}(i) = \frac{v_{\text{lat}}(i)}{\sqrt{\sum_{i=1}^{N} v_{\text{lat}}(i)^2}}, \tag{1}$$

**Table 2.** Simulation experiments performed.

| Year | Experiment Name | Description |
|---|---|---|
| 2008 | 2008_1× | Control simulation |
| | 2008_0× | No emissions from Kilauea |
| | 2008_5× | Five-fold increase in Kilauea emissions |
| | 2008_PH2km | Volcanic plume height increased to 2 km |
| 2018 | 2018_1× | Control simulation |
| | 2018_0× | No emissions from Kilauea |
| | 2018_PH4km | Volcanic plume height increased to 4 km |
| | 2018_PH4kmAsh | 2018_PH4km plus ice nucleation on ash particles |
| 2000–2017 | GEOS$_{CLIM}$ | GEOS climatology excepting 2008 |

where $N$ is the number of latitudes in the domain at 5° intervals (10°–25° N), $v_{lat}(i)$ is the seasonal latitudinal average (JJA 2008 or MJJ 2018) at latitude $i$, and $v_{lat,norm}(i)$ is the seasonal latitudinal anomaly normalized by the latitudinal mean (JJA or MJJ averaged over the climatology) at latitude $i$. To emphasize the location of the plume, we focused on a domain that included only areas west of the source ((25°,–155° NE),(10°,–180° SW)).

## 4  Results and Discussion

The goal of this work was to investigate the role of microphysical processes on ACIs during the Kilauea degassing events. To that end, we first show the reliability of our simulations by comparing satellite retrievals and GEOS control simulations (2008_1× and 2018_1×) and sensitivity experiments (2008_5×, 2008_PH2km, 2018_PH4km, and 2018_PH4km_ash). We then looked for common features in both the 2008 and 2018 Kilauea degassing events and interpreted their differences. Finally, Section 4.4 details the specific microphysical processes involved in cloud modification by the Kilauea emissions. Our results suggested that it is likely that the effects of volcanic emissions on cloud microphysics are specific to cloud phase. When possible, we divide the discussion between effects on ice and liquid clouds.

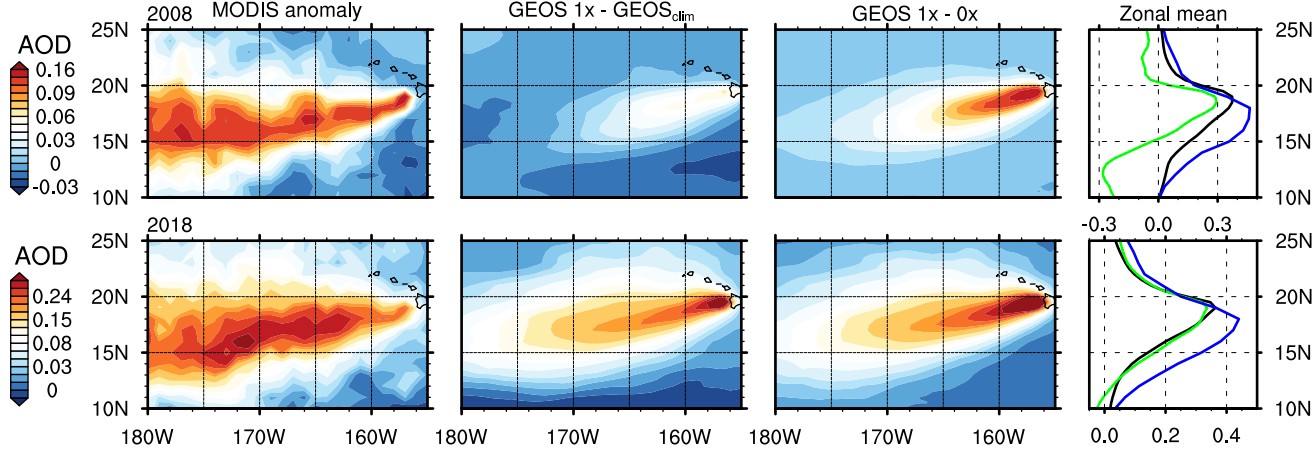

**Figure 2.** AOD anomalies (from left) for MODIS observations and for GEOS simulations using actual emissions (2008_1× and 2018_1×) calculated against GEOS$_{\text{CLIM}}$ and the zero emissions scenarios (2008_0× and 2018_0×), during JJA 2008 (top) and MJJ 2018 (bottom). The rightmost column shows normalized zonal mean anomalies for MODIS (blue), and GEOS against GEOS$_{\text{CLIM}}$ (green) and zero emissions (black).

Anomalies for AOD from MODIS during peak degassing periods are shown in Fig. 2 for JJA 2008 (top left) and MJJ 2018 (bottom left). Figure 2 also shows modeled AOD anomalies for simulations using actual emissions (2008_1×, 2018_1×) calculated against the simulated climatology (GEOS$_{\text{CLIM}}$) and with zero emissions scenarios (2008_0×, 2018_0×). Elevated
aerosol loadings were apparent beginning at Kilauea (19.4° N, 155.2° W) and extended in a near-Gaussian plume westward across the domain shown in Fig. 2. AOD anomalies outside of the plume domain were negligible, supporting the assumption that Kilauea is located in an otherwise "clean" environment, relatively untouched by anthropogenic aerosol emissions from North America or East Asia. During both events, the elevated AOD anomalies observed by MODIS extended westward from Kilauea in the path of the Pacific easterly trade winds (Yuan et al., 2011). This is well reproduced in the GEOS results. AOD
anomalies appear weaker compared to GEOS$_{\text{CLIM}}$ than against 0×, particularly near the Kilauea crater. This is due to the fact that there is always some passive degassing near the source, which would be represented in the climatology but not in the 0× experiment. Passive degassing is shown in Fig. 1, where a thin plume coming from the summit crater was evident during an eruptive pause in 2011, a year without a major volcanic event at the site.

Figure 3 shows MODIS retrievals and GEOS simulations of CF anomalies during the two events. Peak anomalies during
the 2018 event (Fig. 3; bottom) were approximately 2–3× greater than during the 2008 event (Fig. 3; top). In both cases, the 1×−GEOS$_{\text{CLIM}}$ difference (middle panels) reproduced the spatial distribution of the CF anomaly from the MODIS retrieval (within the plume domain as well as zonal means), whereas the 1×−0× differences tended to overestimate the anomaly in 2008 and underestimated it in 2018. In 2008, the normalized 1×−GEOS$_{\text{CLIM}}$ anomaly correlated well with MODIS (Table 4) whereas there was essentially no correlation between the 1×−0× anomaly and the retrieval, indicating that the observed CF
anomaly was mainly driven by meteorologic variability, likely differences in SST (Takahashi and Watanabe, 2016; Boo et al., 2015). Thus ACIs likely had only a minor effect on CF for the two degassing events. It is possible that the aerosol layer may

have locally modified SST, hence indirectly affecting CF. Elucidating this requires coupled ocean-atmosphere simulations and is suggested for future research. The discrepancies between the simulated and MODIS anomalies were relatively larger for CF than for AOD (Figs. 2 and 3). The AOD anomalies are primarily a function of the aerosol load and largely determined by the volcanic events. On the other hand, CF is influenced by many factors including convection, SST, El Niño-Southern Oscillation (ENSO) state, cloud microphysics, and winds, and is much more sensitive to natural variability. Satellite retrievals are also influenced by empirical definitions of cloudy/non-cloudy regions, adding uncertainty (Pincus et al., 2012). Given this, it is remarkable that the CF anomaly against the climatology is in relative good agreement with the satellite retrieval demonstrating the skill of GEOS in reproducing clouds during volcanic events.

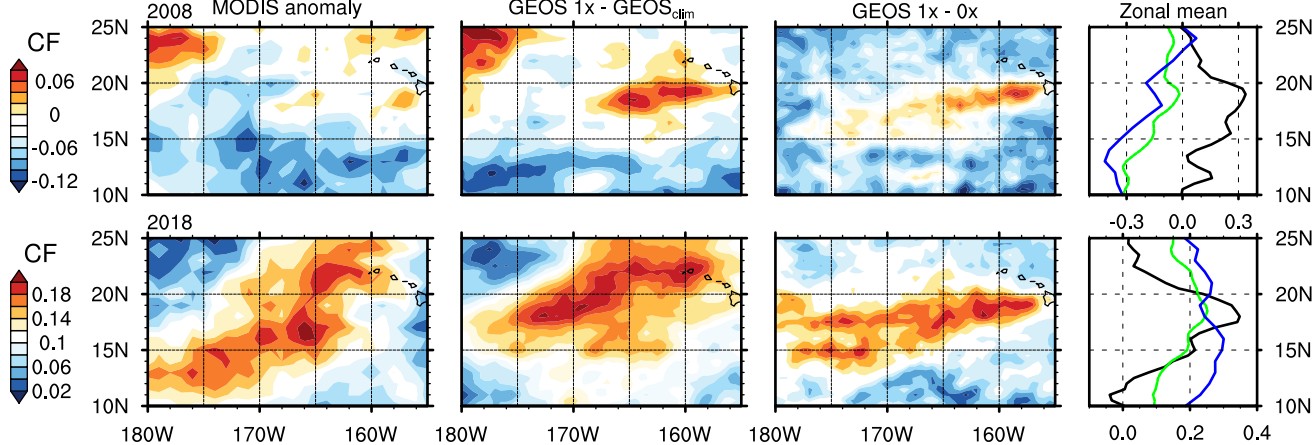

**Figure 3.** Like Fig.2 but showing CF anomalies during JJA 2008 (top) and MJJ 2018 (bottom).

To examine the cloud vertical structure during the two degassing events, GEOS results for the 2008_1× and 2018_1× simulations were compared against CALIPSO-GOCCP, shown in Figs. 4 and 5, respectively. During both events, predominant cloud layers were found at 900 and 200 hPa, corresponding to low-level trade cumulus and *in situ* formation of thin cirrus clouds, respectively. The cloud vertical structure during both events was similar, indicating a strong meteorological control. This was well-captured by the GEOS simulations; the model however tended to overestimate high-level clouds and underestimate low-level clouds, particularly during the 2018 event (Fig. 5). This may result from replaying temperature to the reanalysis (Section 3.2) but letting water vapor evolve freely during the simulations. Such a configuration tends to overestimate relative humidity, particularly at cirrus levels, where the two-moment microphysics allows for supersaturation but MERRA-2 does not, and $T$ is less constrained by observations (Gelaro et al., 2017). The discrepancy may also be exacerbated by artefacts in the retrieval. GOCCP tends to underestimate the presence of thin cirrus clouds, and may overestimate low-level clouds in the presence of high aerosol loading (Chepfer et al., 2010). GEOS was, however, able to simulate an anomalous increase in CF above 800 hPa between 20°N and 25° N in 2008 as seen in the retrieval. Similarly, there was a strong increase in CF in 2018 across the

domain, although GEOS predicted a much larger anomaly effect on cirrus clouds. Again, the discrepancy may be the result of a lack of sensitivity to thin cirrus clouds in GOCCP.

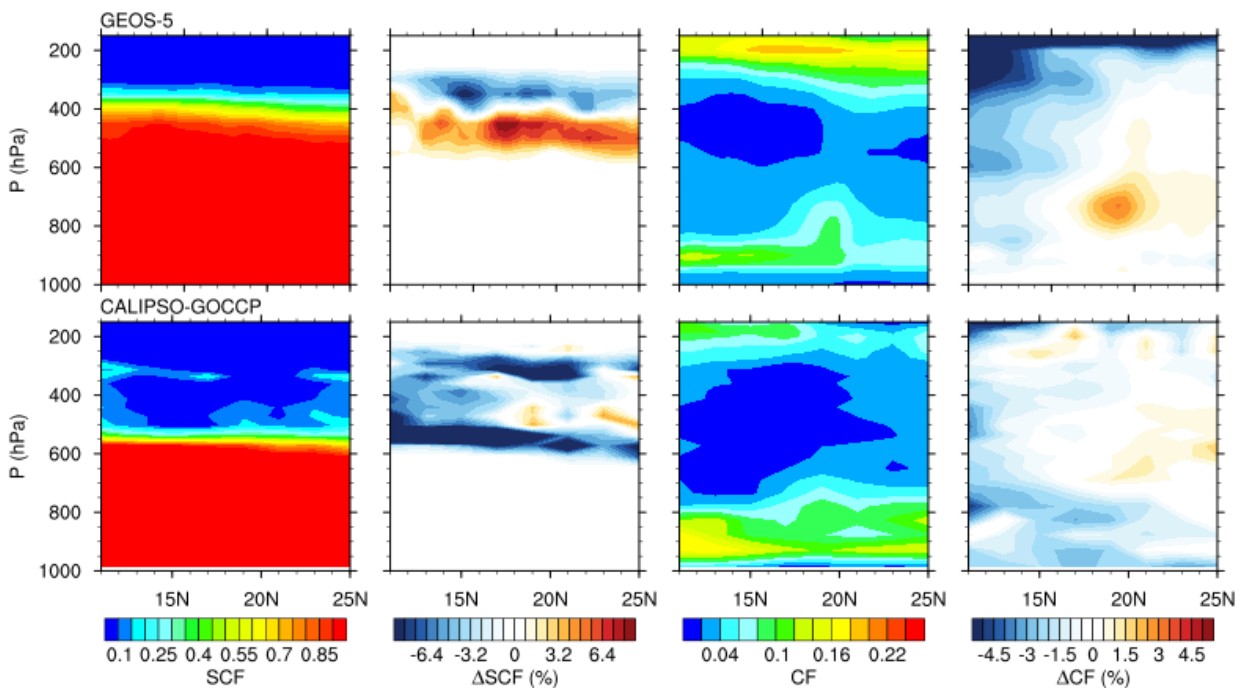

**Figure 4.** Zonal mean anomalies for JJA supercooled cloud fraction ($\Delta$SCF) and cloud fraction ($\Delta$CF) from the GEOS (top) simulation and the CALIPSO-GOCCP (bottom) dataset. Also shown are the anomalies with respect to the 2007–2017 climatology (excluding 2008).

Another interesting feature of Figs. 4 and 5 is the presence of mid-level, alto-cumulus clouds likely maintaining some su-
percooled water. In GOCCP, the SCF (i.e., the fraction of condensate remaining as liquid at $T < 273$ K) decreased sharply at around 550 hPa in both years, but surprisingly remained significant ($> 0.1$) up to 200 hPa where homogeneous ice nucleation glaciated the remaining water. This was particularly true in 2018 where GOCCP showed enhanced SCF relative to the climatology. GEOS showed similar behavior where the supercooled layer ($0 < \text{SCF} < 1$) was deeper in 2018 than in 2008, particularly near the summit crater ($\approx 25°$ N). The supercooled layer was almost $100$ hPa higher in GEOS than in GOCCP, indicating
differences in ice and liquid partitioning between the model and the retrieval, and may be explained by the different definitions of SCF in each case. In GEOS, SCF is calculated on a mass basis, whereas in GOCCP it corresponds to the frequency of pixels that are classified as ice. The positive anomaly in SCF may suggest a deepening of clouds in the presence of aerosol and is analyzed in Section 4.4.

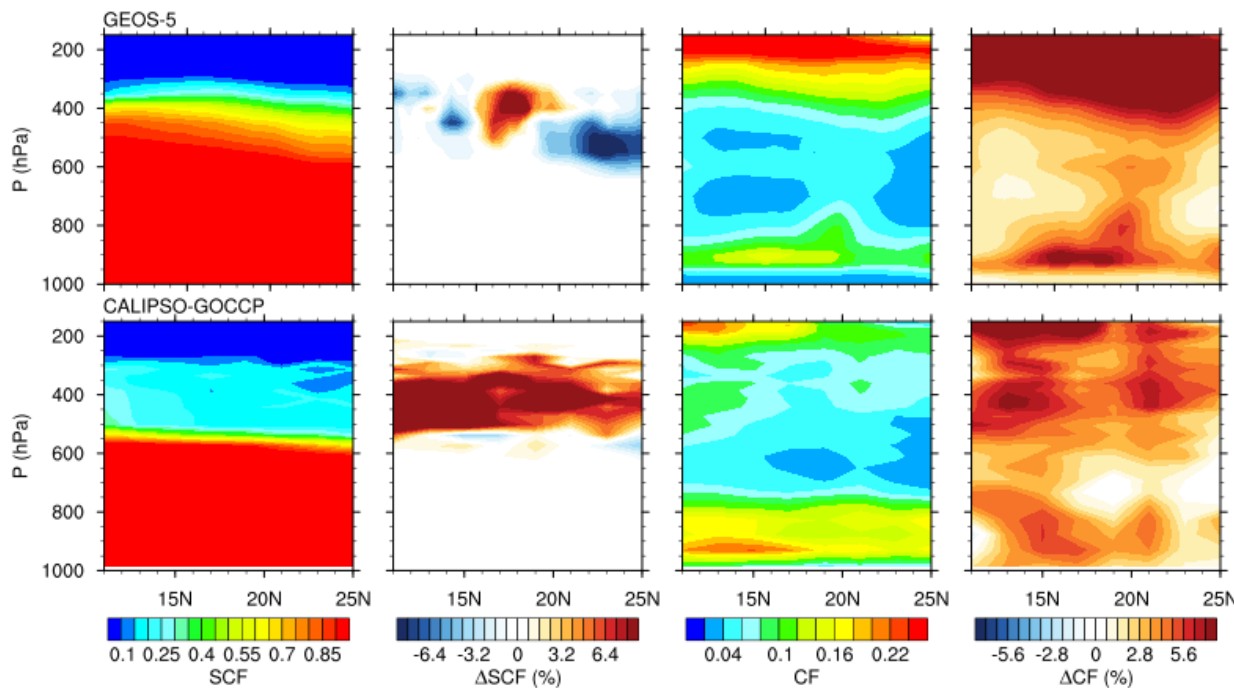

**Figure 5.** Like Figure 4 but for MJJ, 2018.

### 4.1 Liquid Clouds

Figures 6 and 7 show anomalies in $R_{\text{eff}}$, CDNC, COD, and LWP for MODIS and the GEOS control simulations (Table 2) during the 2008 and 2018 degassing events, respectively. Domain mean anomalies are summarized in Table 3. Table 4 shows the coefficient of determination ($R^2$) between the MODIS and the GEOS normalized zonal mean anomalies. During both the 2008 and the 2018 events, GEOS was able to capture the geographical distribution of the anomalies apparent in the MODIS retrievals for liquid clouds. Student's *t*-tests indicated that the anomalies were statistically significant, typically to a 70%–80% level ($p < 0.3$, except for AOD and CF in 2008, which were significant only to 60%) for $R_{\text{eff}}$ and CDNC more so than for other variables (Table 3). For the 2008_1× experiment, the magnitude of the $R_{\text{eff}}$ and CDNC domain-averaged anomalies ($-0.58\,\mu\text{m}$ and $16.04\,\text{cm}^{-3}$, respectively) was in close agreement with MODIS ($-0.75\,\mu\text{m}$ and $21.53\,\text{cm}^{-3}$, respectively), and well-correlated with the satellite retrievals (Table 4). These values are also close to the anomalies calculated against the 2008_0× simulation suggesting a microphysical control to $R_{\text{eff}}$ and CDNC. On the other hand, although the COD and LWP normalized anomalies are well-correlated between GEOS and MODIS, the model overestimated their absolute value. Similarly, in 2018 the GEOS and MODIS anomalies in $R_{\text{eff}}$ ($-1.06\,\mu\text{m}$ vs. $-0.60\,\mu\text{m}$) and CDNC ($26.65\,\text{cm}^{-3}$ vs. $19.43\,\text{cm}^{-3}$) were in better agreement than for COD (7.69 vs. 0.69) and LWP ($15.30\,\text{g\,m}^{-2}$ vs. $5.09\,\text{g\,m}^{-2}$). Normalized zonal mean anomalies were,

however, well-correlated ($R^2 > 0.5$) for CDNC and AOD (Table 4). The discrepancies in LWP and COD absolute anomalies were likely due to variation in SSTs inside the domain, and the lack of a cloud albedo-SST feedback in our simulations.

SO$_2$ emissions at the Mauna Loa Observatory (19.5° N, 155.6° W) peaked in JJA 2008 relative to the 1995–2008 seasonal mean, with prevailing La Niña conditions (Potter et al., 2013). Thus it is likely that the degassing event contributed to lower SSTs, hence lowered surface evaporation rates and LWP. The negative anomaly in LWP in the southernmost part of the domain evident in 2008 is also missing in the 2018 event due to neutral ENSO conditions in the latter (NOAA, 2020), indicating that ENSO exerted a strong meteorological control that could have drown out ACI signatures in the former. Another reason behind
the larger aerosol effects on COD and LWP simulated by GEOS than observed by MODIS may lie in differences in phase partitioning (i.e., liquid vs. ice) (Marchant et al., 2016). To help identify thin cirrus clouds, measured top-of-the-atmosphere (TOA) reflectance at $1.38\,\mu m$ is used to partition high-altitude cirrus clouds from underlying liquid clouds. This method is strongly influenced by the relative humidity of the atmospheric column. Therefore, areas with low column water vapor amount may have more clouds partitioned to the ice phase than is realistic. This is important because 2008 was a La Niña year and
the relative humidity in the atmospheric column was below climatological values, so LWP anomalies may appear low due to enhanced partitioning to the ice phase in the MODIS cloud phase classification algorithm. Even in 2018, most of the total water path (TWP) anomaly resulted from an increase in the ice water path, indicating strong partitioning to the ice phase in the MODIS retrieval (Table 4).

        Overall, more robust correlations ($R^2 > 0.5$) resulted when the anomalies were calculated against the GEOS$_{CLIM}$ than against
the no-emissions scenarios (Table 4). This indicated that while both GEOS climatology and 0× cases included aerosol loading consistent with MODIS retrievals (Fig. 2, Table 4), correlations with observed data for liquid clouds were sensitive to meteorological effects captured by the MODIS and GEOS climatologies. MODIS TWP anomalies (where TWP is the sum of LWP and IWP) in 2018 were about 20× higher than those reported for the 2008 event (Table 3), suggesting that ACIs for 2018 were not limited to liquid clouds.

We found that effects on liquid clouds were more statistically significant in 2018 than in 2008 for both observed (MODIS) and simulated (GEOS) cloud properties. In 2018, the magnitude and significance level of the simulated anomalies against GEOS$_{CLIM}$ were close to those calculated against 2018_0× indicating that increased aerosol loading, regardless of injection height, resulted in the heightened development of liquid cloud droplets, and that these effects could be attributed to volcanic degassing as opposed to regional meteorology. The results of the 2008_5× run were in general highly statistically significant
($p < 0.05$), and agreed in magnitudes with the 2018_1× experiment. The similarities in the magnitudes of simulated anomalies for 2008_5× and 2018_1× suggested that increased aerosol loadings would have been sufficient to overcome meteorological effects which dampened the 2008 JJA anomalies with respect to long-term behavior. This and the similarities in spatial patterns (i. e., anomalies largely constrained by and maximized within the plume domain) for cloud anomalies in JJA 2008 (Fig. 6) and MJJ 2018 (Fig. 7) suggested a threshold response to overcome meteorological effects that was largely controlled by emissions.

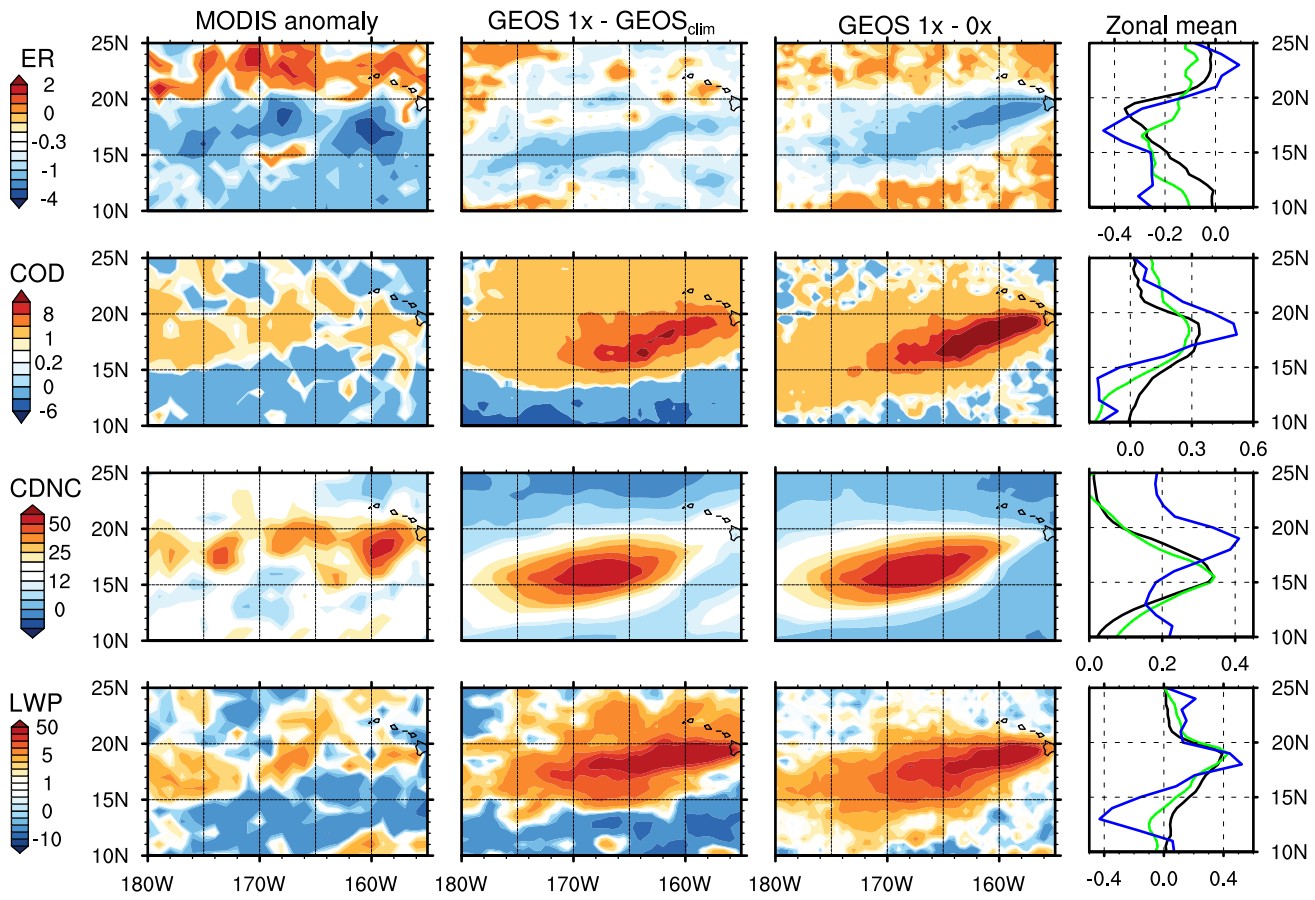

**Figure 6.** Anomaly in liquid cloud properties during JJA 2008 for (from top) $R_{\text{eff}}$ (μm, shown as ER), COD, CDNC (m$^{-3}$), and LWP (g m$^{-2}$). Anomalies are shown (from left) for MODIS observations, and for GEOS simulations calculated against GEOS$_{\text{CLIM}}$ and the zero emissions scenario. The rightmost column shows normalized zonal mean anomalies for MODIS (blue), and GEOS against GEOS$_{\text{CLIM}}$ (green) and zero emissions (black).

**Table 3.** Anomalies for liquid cloud properties, and for AOD and CF. Values in parentheses are the $p$-values associated with a one-sided Student's $t$-test against the long-term mean.

| Experiment | Exp vs. climatology (excluding 2008) | | | | | | | Exp vs. 0× | | | | | | |
|---|---|---|---|---|---|---|---|---|---|---|---|---|---|---|
| | $R_{\text{eff}}$ (μm) | COD (–) | CDNC (m$^{-3}$) | LWP (g m$^2$) | TWP (g m$^2$) | AOD (–) | CF (–) | $R_{\text{eff}}$ (μm) | COD (–) | CDNC (m$^{-3}$) | LWP (g m$^2$) | TWP (g m$^2$) | AOD (–) | CF (–) |
| **2008_1×** | -0.58 (0.08) | 4.63 (0.12) | 16.04 (0.23) | 8.77 (0.19) | 8.31 (0.15) | 0.01 (0.40) | -0.03 (0.41) | -0.65 (0.07) | 3.77 (0.17) | 15.68 (0.24) | 6.82 (0.27) | 7.17 (0.19) | 0.03 (0.07) | 0.02 (0.63) |
| **2008_5×** | -1.31 (0.01) | 10.70 (0.02) | 40.06 (0.01) | 18.99 (0.05) | 19.34 (0.02) | 0.07 (0.07) | -0.01 (0.78) | -1.37 (0.01) | 9.84 (0.02) | 39.71 (0.01) | 17.04 (0.06) | 18.21 (0.02) | 0.09 (0.04) | 0.03 (0.32) |
| **2008_PH2km** | -0.58 (0.08) | 4.63 (0.12) | 16.04 (0.23) | 8.77 (0.19) | 8.70 (0.13) | 0.01 (0.40) | -0.03 (0.41) | -0.45 (0.07) | 2.31 (0.17) | 13.55 (0.24) | 4.03 (0.27) | 4.90 (0.17) | 0.02 (0.07) | 0.01 (0.63) |
| **MODIS (2008)** | -0.75 (0.26) | 0.46 (0.18) | 21.53 (1.00) | 0.65 (0.52) | -0.89 (0.74) | 0.03 (0.04) | -0.02 (0.50) | – (–) | – (–) | – (–) | – (–) | – (–) | – (–) | – (–) |
| **2018_1×** | -1.06 (0.15) | 7.69 (0.13) | 26.65 (0.10) | 15.30 (0.08) | 45.71 (0.07) | 0.10 (0.16) | 0.15 (0.18) | -1.23 (0.11) | 8.30 (0.12) | 28.61 (0.08) | 10.07 (0.16) | 18.09 (0.28) | 0.10 (0.15) | 0.02 (0.78) |
| **MODIS (2018)** | -0.60 (0.14) | 0.69 (0.24) | 19.43 (0.02) | 5.09 (0.21) | 25.81 (0.23) | 0.11 (0.00) | 0.13 (0.20) | – (–) | – (–) | – (–) | – (–) | – (–) | – (–) | – (–) |

**Table 4.** $R^2$ values for the comparison of GEOS normalized zonal anomalies for liquid clouds against MODIS.

| Experiment | MODIS anomaly vs GEOS 1×−clim | | | | | | | MODIS anomaly vs GEOS 1×−0× | | | | | | |
|---|---|---|---|---|---|---|---|---|---|---|---|---|---|---|
| | $R_{\text{eff}}$ (μm) | COD (–) | CDNC (m$^{-3}$) | LWP (g m$^2$) | TWP (g m$^2$) | AOD (–) | CF (–) | $R_{\text{eff}}$ (μm) | COD (–) | CDNC (m$^{-3}$) | LWP (g m$^2$) | TWP (g m$^2$) | AOD (–) | CF (–) |
| **2008_1×** | 0.49 | 0.70 | 0.30 | 0.57 | 0.68 | 0.66 | 0.68 | 0.45 | 0.45 | 0.40 | 0.26 | 0.02 | 0.88 | 0.01 |
| **2008_PH2km** | 0.49 | 0.70 | 0.30 | 0.57 | 0.65 | 0.66 | 0.68 | 0.45 | 0.45 | 0.40 | 0.26 | 0.01 | 0.88 | 0.01 |
| **2018_1×** | 0.01 | 0.31 | 0.67 | 0.58 | 0.24 | 0.91 | 0.13 | 0.23 | 0.33 | 0.83 | 0.41 | 0.22 | 0.86 | 0.28 |

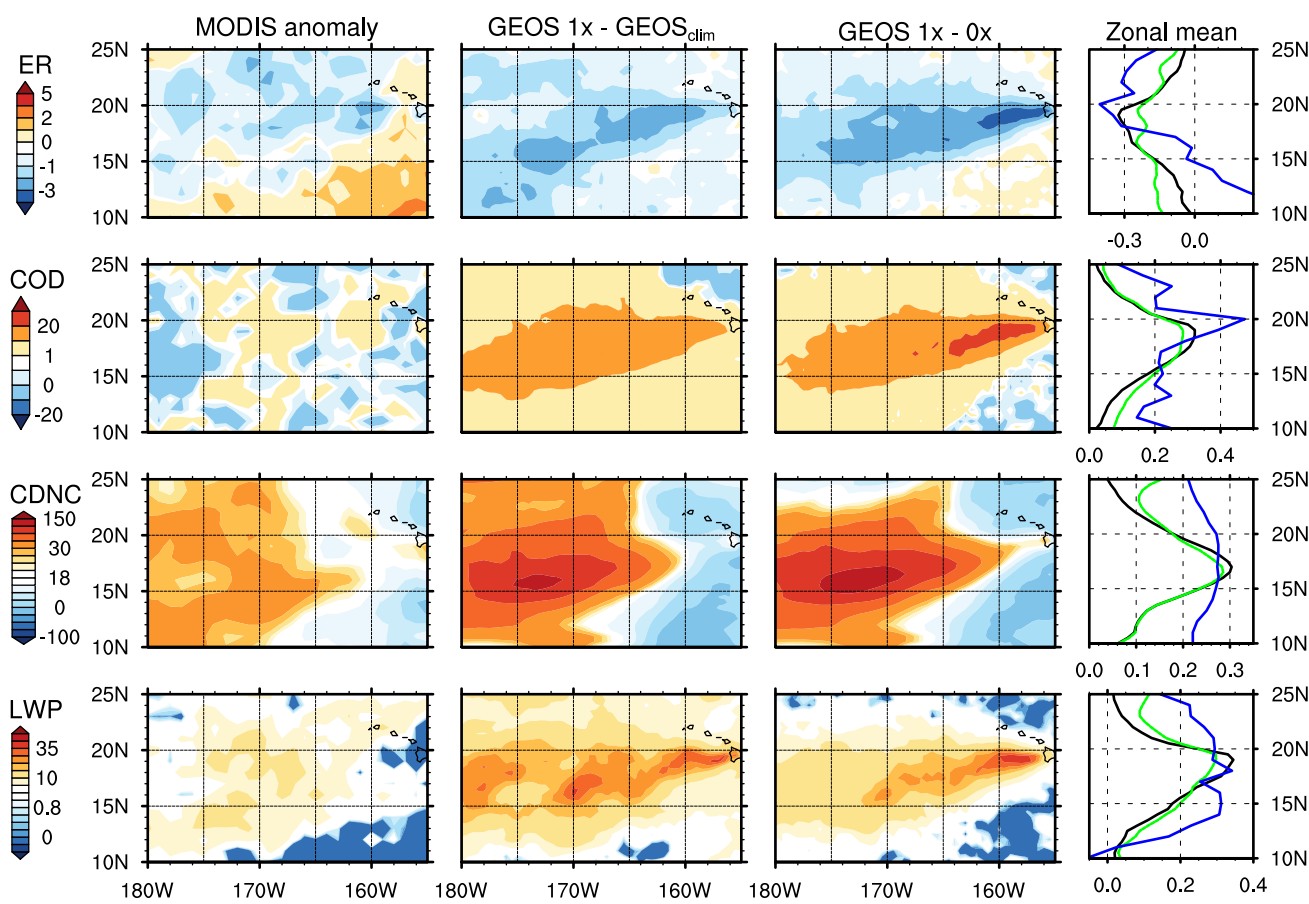

**Figure 7.** Like Fig. 6 but for MJJ 2018.

## 4.2 Ice Clouds

The volcanic plume height during peak degassing periods varied significantly between 2008 and 2018, therefore we expected the strength of ACIs for ice clouds to be partially dependent on the availability of INPs. It is likely that the plume introduced $SO_2$ and ash particles to the upper troposphere - directly in 2018 and via convection in 2008. The effects of INPs on ice cloud development depend strongly on the microphysical processes dominating cloud evolution as well as the efficiency with which the new particles can promote ice crystal formation (Barahona et al., 2010). Additionally, the presence of aerosols in the upper troposphere was not apparent in a Gaussian-like plume emanating from Kilauea Volcano, as was the case for liquid clouds, therefore the identification of ACIs for ice clouds was less straight-forward.

Figure 8 shows that during the 2008 event, the aerosol plume did not significantly alter the properties of ice clouds. The 2008_1×−GEOS$_{CLIM}$ anomalies in ice clouds appeared spatially consistent with MODIS anomalies for COD and IWP, whereas the 2008_1×−0× showed little to no variability in the domain. As shown in Table 6, there was high correlation between MODIS and the 2008_1×−GEOS$_{CLIM}$ normalized zonal mean anomalies for COD ($R^2 = 0.84$) and IWP ($R^2 = 0.93$). These correlations substantially decreased when considering comparisons between MODIS and the 2008_1×−2008_0× difference. For COD in 2008, the latter seemed to retain some correlation ($R^2 = 0.29$), and may indicate some level of microphysical control on the observed anomalies. This suggested that most of the observed ice cloud anomaly in 2008 resulted from meteorological and SST variability. Prevailing La Niña conditions in 2008 likely reduced relative humidity within the atmospheric column, particularly in the upper troposphere, thereby limiting the supply of water vapor necessary for ice crystal nucleation and growth. Anomalous conditions in ice clouds with respect to climatological and background conditions, however, were not statistically significant (Table 5).

Similar to the 2008 event, there seemed to be a strong meteorological component during MJJ 2018 degassing because the spatial correlation of the 2018_1×−GEOS$_{CLIM}$ results appeared consistent with MODIS anomalies (Fig. 9). Additionally, correlations between MODIS and GEOS$_{CLIM}$ zonal mean anomalies were higher ($R^2 \geq 0.2$) than for the MODIS vs. 2018_1×−2018_0× correlations, with the exception of IWP ($R^2 = 0.32$) (Table 6). Unlike the 2008 event, the 2018_1×−2018_0× anomalies showed variability within the domain that was of similar magnitude to and appeared spatially consistent with GEOS$_{CLIM}$ anomalies (Fig. 9), indicating that not all ACIs in ice clouds could be attributed to meteorology alone. Figure 9 shows similar spatial distributions for observed and simulated anomalies within the plume domain, in particular between $175°$ W and $165°$ W. Whereas for $R_{eff}$ and COD anomalous cloud properties appeared to be related to meteorological variability, IWP anomalies appeared to have a strong microphysical component. The MODIS and GEOS$_{CLIM}$ anomalies for IWP in 2018 were of the same order of magnitude, $20.59\,\mathrm{g\,m^{-2}}$ and $30.41\,\mathrm{g\,m^{-2}}$, respectively (Table 5), whereas the GEOS IWP 2018_1×−2018_0× difference was $\approx 8.03\,\mathrm{g\,m^{-2}}$. Although the strength of the observed effect was better represented by the GEOS$_{CLIM}$ IWP anomaly, the IWP correlation for MODIS vs. 2018_1×−2018_0× was slightly higher than against the climatology ($R^2 = 0.32$ vs. $R^2 = 0.23$); we found this difference to be significant at the $95\%$ confidence level. This strongly suggested a significant microphysical control on the IWP anomaly during the 2018 event.

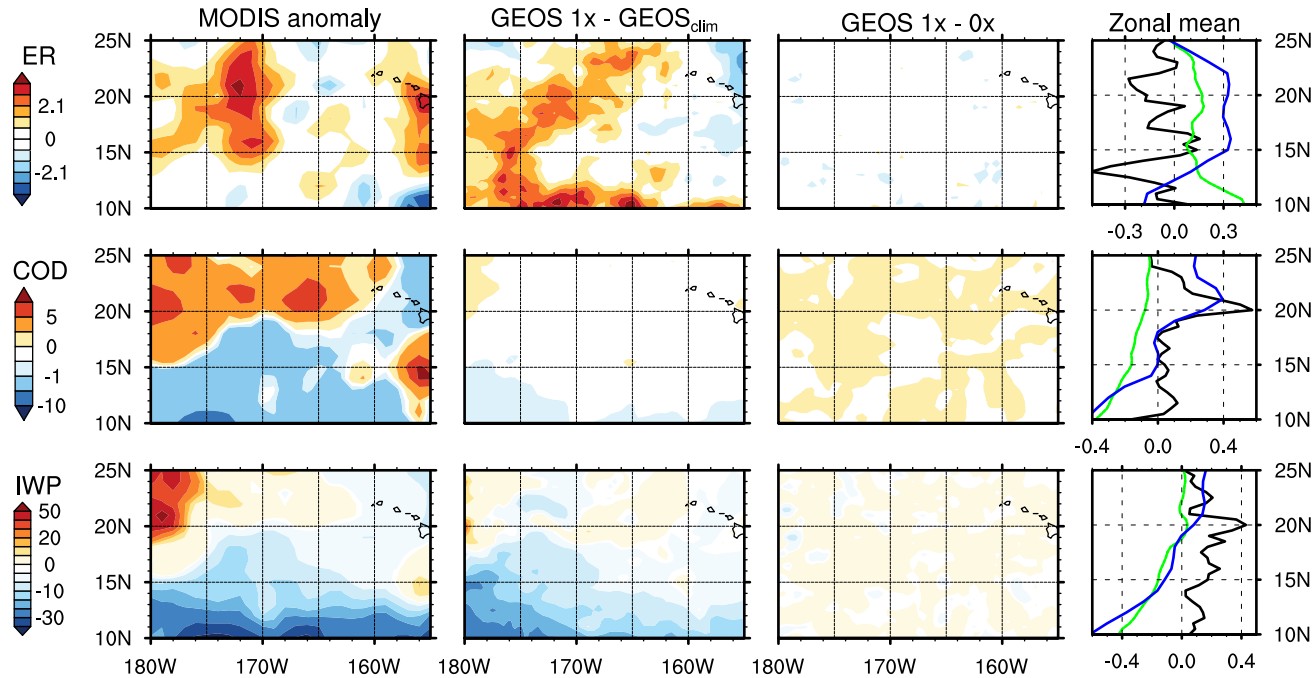

**Figure 8.** Ice cloud anomalies during JJA 2008 for (from top) $R_{\text{eff}}$ (μm, shown as ER), COD (–), and IWP (g m$^2$). Anomalies are shown (from left) for MODIS observations, and for GEOS simulations calculated against GEOS$_{\text{CLIM}}$ and the zero emissions scenario. The rightmost column shows normalized zonal mean anomalies for MODIS (blue), and GEOS against GEOS$_{\text{CLIM}}$ (green) and zero emissions (black).

## 4.3 Role of Injection Height and Ash Content

Domain-averaged CDNC varied widely between the different experiments, except for the cases with elevated plume heights
(2008_PH2km and 2018_PH4km), which essentially overlapped with the 2008_1× and 2018_1× experiments, respectively (Figs. 10 and 11). Despite this, increasing the plume height yielded sizable differences in some of the anomalies, particularly for ice clouds (Table 5) and for the 2018 event, although the effect was small. Similarities between MODIS and GEOS results for the 2008_5× and 2018_1× experiments on liquid clouds (Table 3) indicated that increased aerosol loadings during the 2008 event mimicked effects on liquid cloud formation during the 2018 event, most notably for CDNC and AOD. This showed
that liquid cloud sensitivity to the first AIE was dominated by aerosol loadings as opposed to plume morphology during the 2008 event. Increasing plume height in concert with aerosol loadings exceeding 20 kt SO$_2$ day$^{-1}$ (2018_PH4km) reduced the anomaly in IWP and COD by less than 5% relative to the 2018_1× experiment (Table 6). Similarly, accounting for ash INP (2018_PH4km_ash) slightly increased the COD and IWP anomalies (Table 5). Although it had opposite effects in ice crystal mixing ratio, $Q_{\text{ice}}$, and ICNC, with the former lower, and the latter higher than in the 2018_PH4km experiment (Figure 5).
This is further analyzed in Section 4.4. The correlations between GEOS anomalies and MODIS for $R_{\text{eff}}$ and COD in ice clouds improved slightly by the inclusion of ash INP (Table 6).

**Table 5.** Anomalies for ice cloud properties. Values in parentheses are the *p*-values associated with a one-sided Student's *t*-test against the long-term mean.

| Experiment | Exp vs. climatology | | | Exp vs. 0× | | |
|---|---|---|---|---|---|---|
| | $R_{\mathrm{eff}}$ (µm) | COD (–) | IWP (g m$^2$) | $R_{\mathrm{eff}}$ (µm) | COD (–) | IWP (g m$^2$) |
| **2008_1×** | 0.25 (0.73) | -0.13 (0.19) | -0.46 (0.80) | -0.04 (0.96) | 0.00 (0.99) | 0.35 (0.85) |
| **MODIS (2008)** | 0.53 (0.60) | 0.79 (0.40) | -1.46 (0.55) | – (–) | – (–) | – (–) |
| **2018_1×** | 0.95 (0.39) | 0.58 (0.21) | 30.41 (0.16) | -0.09 (0.92) | 0.04 (0.91) | 8.03 (0.63) |
| **2018_PH4km** | 1.12 (0.32) | 0.58 (0.20) | 30.38 (0.16) | 0.08 (0.94) | 0.04 (0.90) | 8.00 (0.62) |
| **2018_PH4km_ash** | -4.33 (0.01) | 0.63 (0.18) | 36.44 (0.15) | -5.38 (0.01) | 0.10 (0.79) | 14.06 (0.47) |
| **MODIS (2018)** | 0.00 (0.27) | 3.03 (0.18) | 20.59 (0.26) | – (–) | – (–) | – (–) |

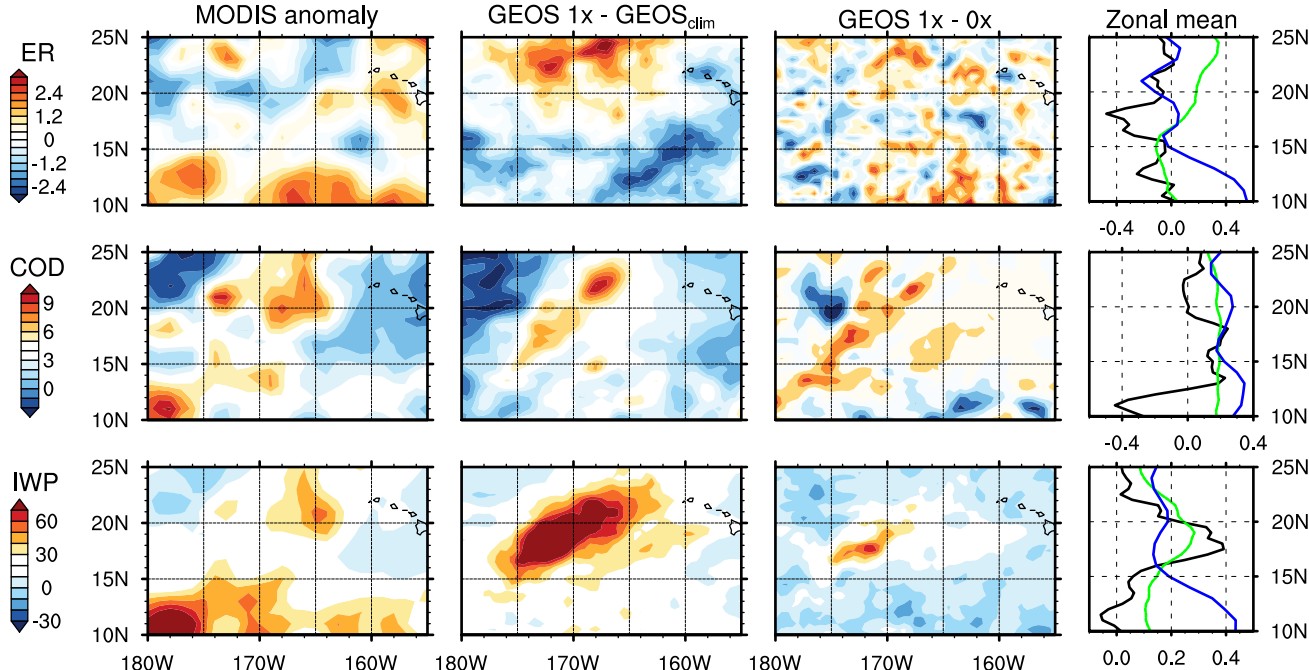

**Figure 9.** Like Fig.8 but for MJJ 2018

**Table 6.** $R^2$ values comparing GEOS normalized zonal anomalies for ice clouds against MODIS.

| Experiment | MODIS anomaly vs GEOS 1×−clim | | | MODIS anomaly vs GEOS 1× vs. 0× | | |
|---|---|---|---|---|---|---|
| | $R_{\mathrm{eff}}$ (μm) | COD (−) | IWP (g m$^2$) | $R_{\mathrm{eff}}$ (μm) | COD (−) | IWP (g m$^2$) |
| **2008_1×** | 0.25 | 0.84 | 0.93 | 0.01 | 0.29 | 0.04 |
| **2018_1×** | 0.23 | 0.15 | 0.23 | 0.03 | 0.16 | 0.32 |
| **2018_PH4km** | 0.25 | 0.16 | 0.20 | 0.14 | 0.04 | 0.34 |
| **2018_PH4km_ash** | 0.27 | 0.39 | 0.15 | 0.29 | 0.01 | 0.21 |

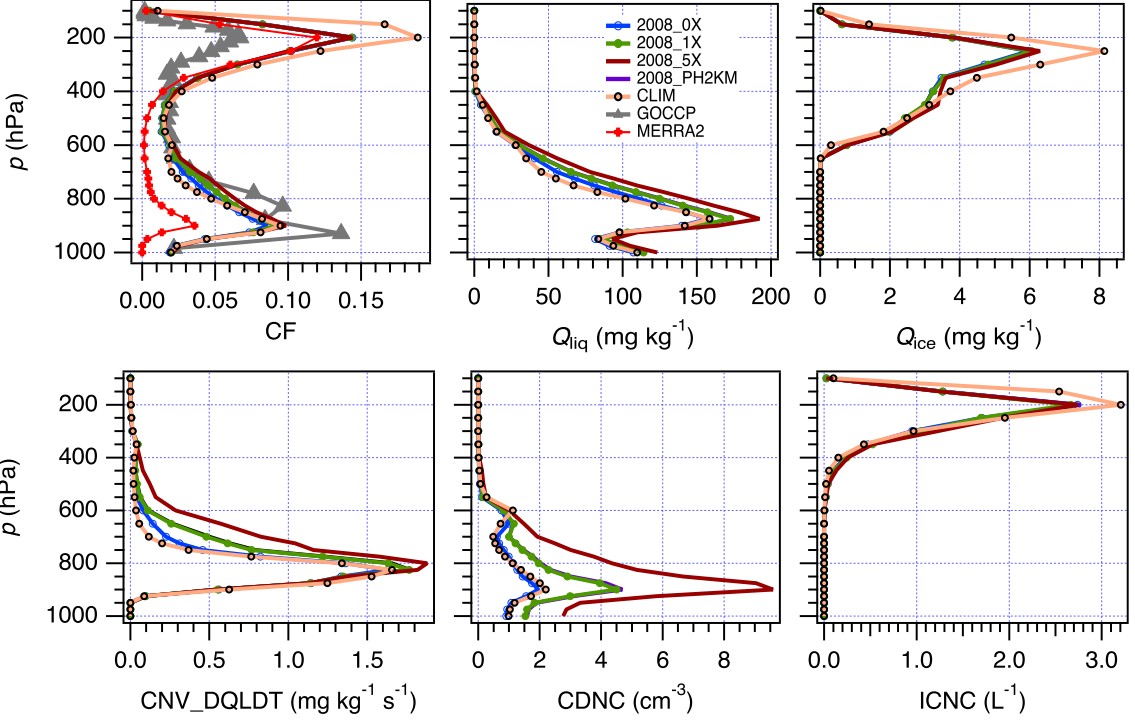

**Figure 10.** Zonal mean vertical profiles for cloud fraction (CF), liquid ($Q_{\mathrm{liq}}$) and ice ($Q_{\mathrm{ice}}$) mixing ratios, total detrained condensate tendency (CNV_DQLDT), cloud droplet (CDNC) and ice crystal (ICNC) number concentration, for the 2008 JJA season. Notice that MERRA2 and GOCCP data are included for CF only.

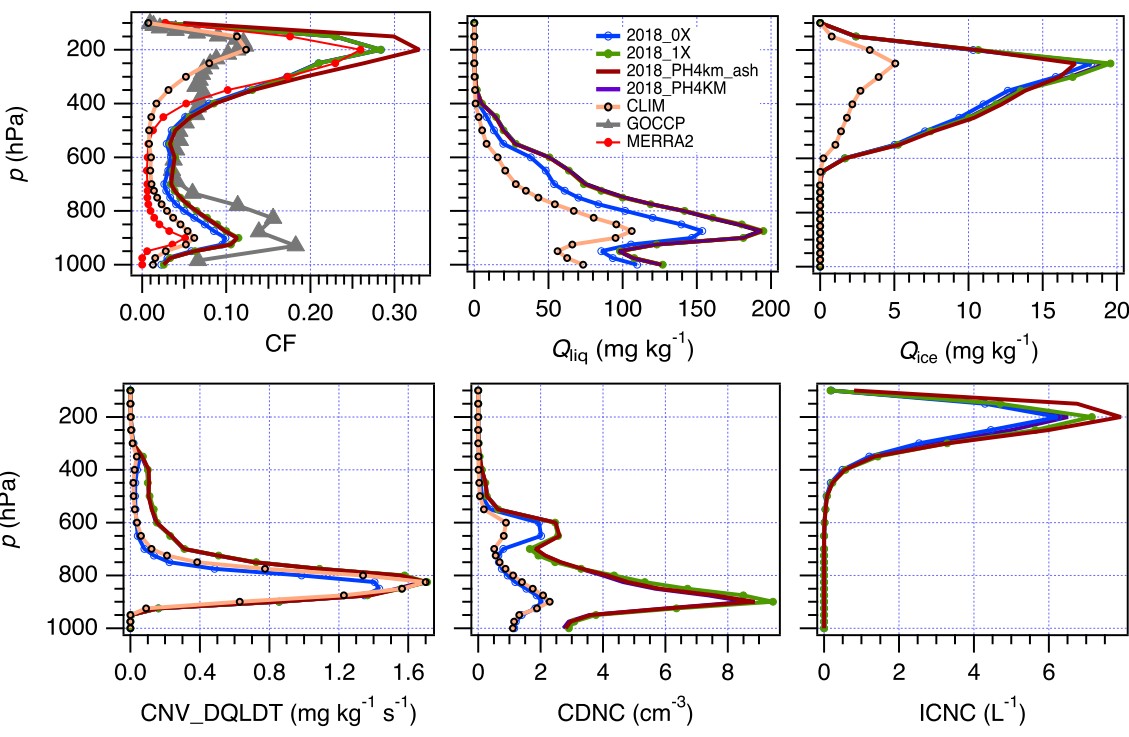

**Figure 11.** Equivalent to Figure 10 but for the 2018 MJJ season.

## 4.4  Microphysical Controls on ACIs

Figures 10 and 11 summarize the experiments performed in this work. Also shown are the CF from MERRA-2 and GOCCP. In all runs, and for both events, the vertical profile of CF was remarkably similar with two predominant cloud layers associated with shallow cumulus and warm cirrus/altocumulus. This vertical structure was in agreement with GOCCP data (also shown in Figs. 4 and 5). Compared to GOCCP, GEOS tended to overestimate high-level CF and underestimated it near the surface. This was discussed in Section 4, however here we also note that there is a high uncertainty in the retrieval during these periods. For example, GEOS-simulated high-level clouds were in better agreement with MERRA-2, whereas low-level CF in the latter was much lower than indicated by GOCCP. Examination of the GEOS simulations suggested that CF around 800 hPa was slightly higher for high aerosol loading than for the no emissions experiments indicating deepening of clouds from the injection of aerosol. This was supported by an increase in $Q_{\text{liq}}$ between the 1× and the 0× experiments and runs along an enhanced detrained mass tendency, CNV_DQLDT, indicating predominant convective effects. In general, the GEOS climatology (CLIM) tended to show lower values of liquid and ice properties than the 0× experiments, with the notable exception of the ice mixing ratio and number concentration in 2008.

These effects can be understood in light of the modification to dominant microphysical tendencies by elevated aerosol emissions. Figures 12 and 13 show the seasonally averaged rates of different cloud microphysical processes that took place during each event. They are distinguished by processes that affect mass and number concentration of liquid droplets and ice crystals. Both events showed similar features regarding processes affecting liquid condensate and CDNC (panels (a) and (c) in Figs. 12 and 13). As expected, liquid cloud microphysics was dominated by CCN activation and condensation (i.e., by droplet formation on sulfate particles). Cumulus detrainment (DCNVL) is a significant source of condensate but only plays a minor role in determining CDNC. The main sinks of liquid mass and number concentration are droplet autoconversion (AUT) and accretion of cloud droplets by rain (ACRL_RAIN). The CCN source rate in 2018 (Fig. 13 c) was about 3× that of 2008 (Fig. 12); however, the liquid condensate tendencies, panel (a) in both figures, were of the same magnitude. Thus the mechanism for the decrease in $R_{\text{eff}}$ was an increase in CCN activation, hence CDNC, which did not translate into a proportional increase in liquid mass (i. e., the first AIE). This is revealed in Figs. 10 and 11 as large increases in CDNC, but only slight increases in $Q_{\text{liq}}$, between the 1× and the 0× experiments.

It is not clear why the liquid autoconversion tendencies (AUT) were almost insensitive to the aerosol load. Most likely, the liquid mass and number concentration sinks were controlled by accretion rather than autoconversion. This is depicted in Fig. 14. Panels (a) and (c) show that the autoconversion sinks for mass and number were similar across all simulation experiments, even those without volcanic emissions (i.e., 2008_0× and 2018_0×), despite CCN activation tendencies varying by a factor of 4. Our results indicated that ACRL_RAIN was, however, enhanced for GEOS experiments with high aerosol concentrations (Fig. 14 c). This suggested an unexpected ACI mechanism. Ingestion of CCN within convective parcels inhibits the formation of precipitation in cumulus clouds; hence more condensate was detrained to the top. DCNVL was thus enhanced in simulations with high aerosol loading (2018_1×, 2008_1×, and 2008_5× in Fig. 14 a). This was reflected in the vertical profiles of CNV_DQLDT, $Q_{\text{liq}}$, and to a lesser extent in CF (Figs. 10 and 11). This mechanism created deeper clouds that

precipitated from above scavenging the liquid below, which explained the increase in ACRL_RAIN as the CCN activation tendency increased. Because droplets also freeze higher in the convective parcel, it may also lead to convective invigoration, although our setup (i.e., replaying $T$ to the reanalysis) prevented GEOS from explicitly simulating this. These findings are consistent with the deepening of the cloud layers described by Yuan et al. (2011) and the evidence for convective invigoration

reported by Mace and Abernathy (2016). They also explain the anomalous enhancement of the mid-level CF and the increased SCF during the two events found in the GOCCP retrievals (Figs. 4 and 5).

Whereas a few processes dominate the microphysics of liquid water, ice microphysics is much more complex. It is clear from Figs. 12 and 13 that ice nucleation (ICENUC), sedimentation (SDM), and growth/sublimation (DEP) were dominant. However other processes, notably convective detrainment (DCNVI) and the Weneger-Bergeron-Findeisen (WBF) process, still

played significant roles in cloud development. Ice autoconversion to snow (AUTICE) was a significant sink for ice mass but not for number concentration. It is likely that the latter was controlled by SDM. This complex microphysical make up may be one of the reasons why the anomalies for ice clouds were much less evident than for liquid clouds, buffering the former against aerosol perturbations.

It is also interesting that almost every ice tendency was 2–3× larger in the 2018 (Fig. 13) than in the 2008 event (Fig. 12),

while the liquid condensate rates were of the same magnitude across all experiments. The main reason for this was that $Q_{\text{ice}}$ was larger in 2018 than in 2008 (Figs. 10 and 11), likely due to higher SSTs during neutral ENSO conditions relative to La Niña SST cooling in 2008. Warmer SSTs triggered more frequent convection events and enhanced the transport of condensate, water vapor, and frozen condensate to the upper troposphere. Crystals sedimenting from above find a favorable environment to grow (which also enhances splintering processes below, represented by ice splintering (HM) in Fig. 13). Although there

was definitely a higher rate for ICENUC in 2018 than in 2008 (Fig. 14), the fact that it changed little upon modification of the injection height or by considering ash as INP, strongly suggested that changes in the vertical transport of liquid and water vapor, rather than ICENUC, led to the observed anomalies in ice clouds. In fact, accounting for ash INP had a negligible effect on immersion freezing rates (not shown). Including ash INP led to slightly higher ice nucleation rates between 200– 300 hPa. Because these crystals grow and sediment quickly, these may explain the slightly lower $Q_{\text{ice}}$ values with respect to

the 2018_1× experiment despite the higher ICNC (Fig. 11).

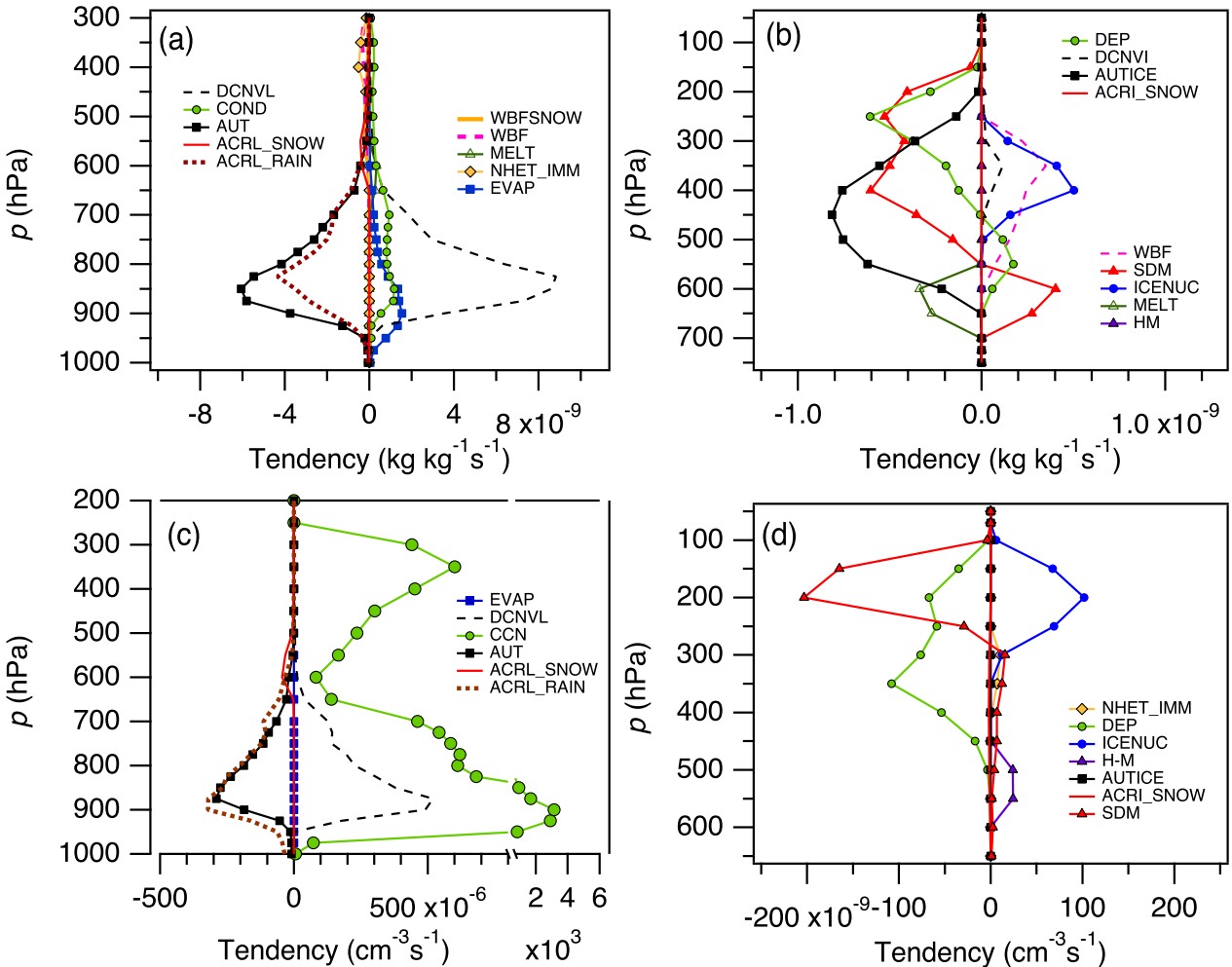

**Figure 12.** Domain-averaged cloud microphysical tendencies for the 2008 JJA season, for (a) liquid mass, (b) ice mass, (c) liquid number, and (d) ice number concentration. Shown are Bergeron-Findeinsen process on ice (WBF) and snow (WBFSNOW), melting (MELT), total ice nucleation, and by immersion freezing only (ICENUC and NHET_IMM, respectively), droplet evaporation (EVAP), convective detrainment of liquid and ice (DCNVL and DCNVI), condensation (COND), liquid and ice autoconversion (AUT and AUTICE, respectively), accretion of liquid by rain and snow (ACRS and ACRL, respectively), CCN activation (CCN), ice sedimentation (SDM), ice crystal growth and accretion by snow (DEP, ACRI_SNOW, respectively), and ice splintering (HM).

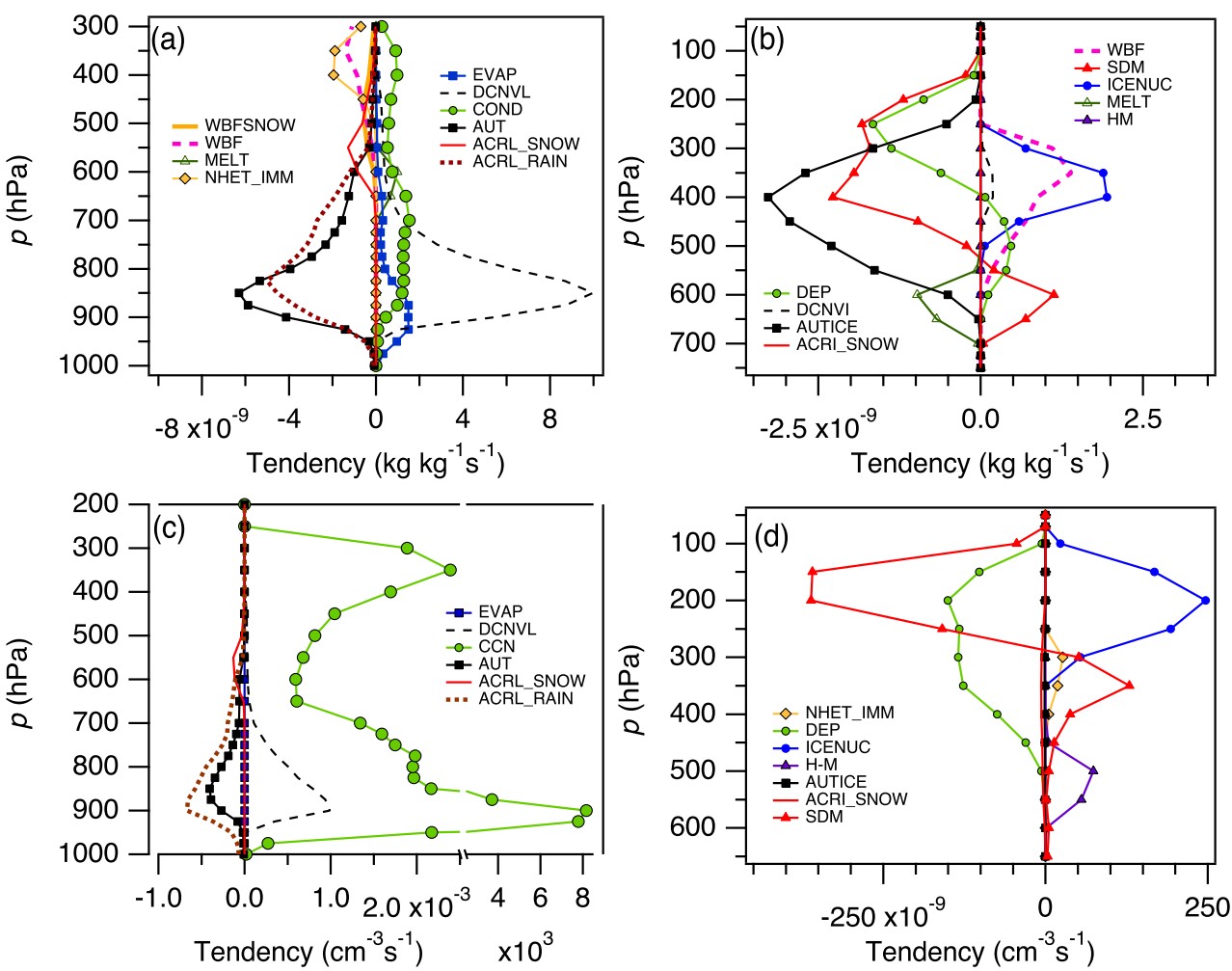

**Figure 13.** Like Fig. 12 but for the 2018 MJJ season.

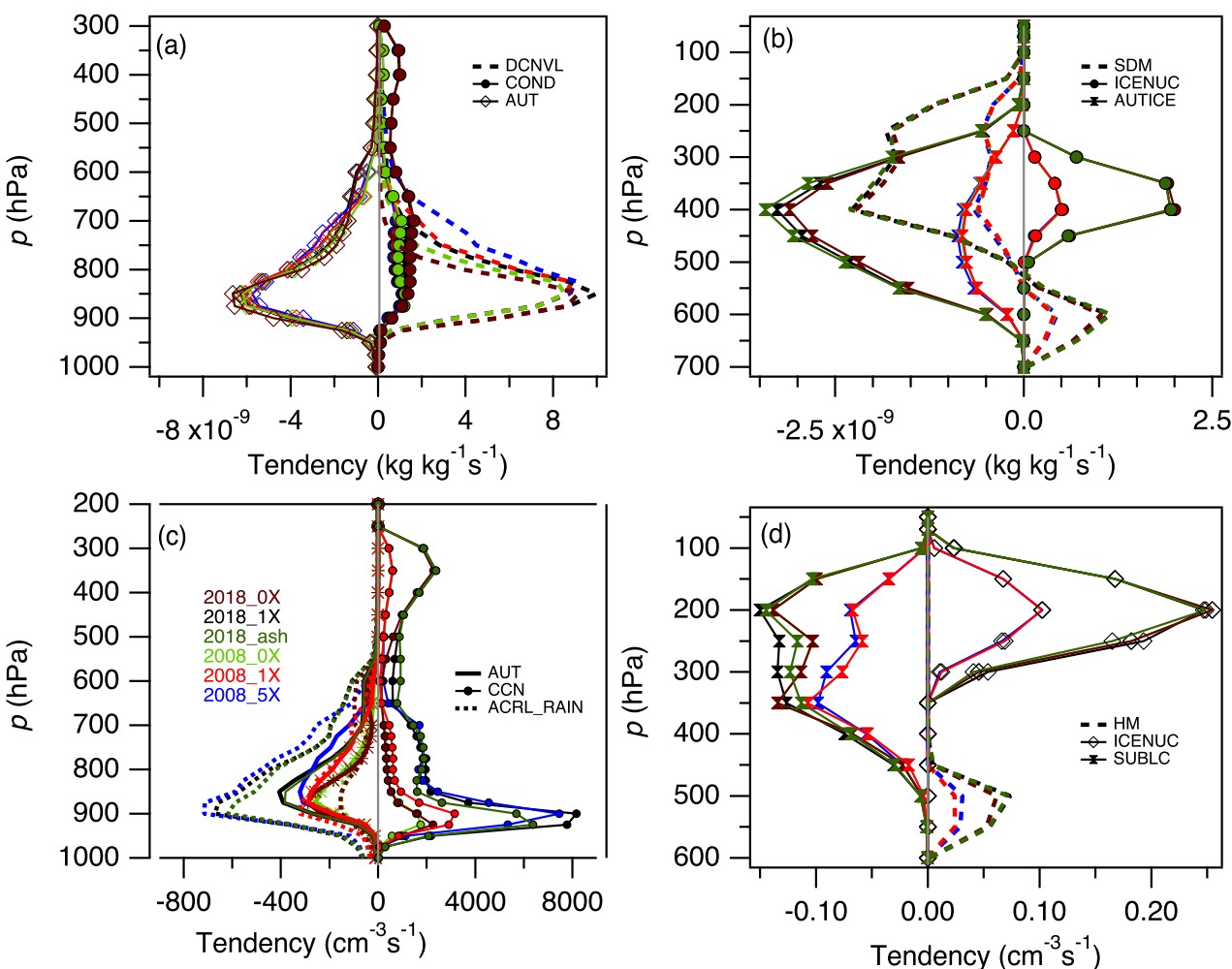

**Figure 14.** Comparison of dominant microphysical process rates for (a) liquid mass, (b) ice mass, (c) liquid number, and (d) ice number concentration. Color corresponds to the the different experiments listed in Table 2.

## 5  Conclusions

We carried out a comprehensive analysis of the effects of volcanic emissions on cloud formation and microphysical properties during two major events from the Kilauea summit crater, Halemaumau, in the Hawaiian islands, occurring during the late spring and summer of 2008 and 2018. Previous analyses characterized the geologic and atmospheric effects of the 2008 event, but this is the first time that such an analysis has been performed for the 2018 event. We combined satellite data, reanalysis products, and atmospheric modeling simulations to analyze and understand the role of meteorology and aerosol-cloud interactions on cloud evolution.

Several factors contributed to observed and simulated similarities/differences for each event: (1) aerosol loading, (2) plume height and composition, and (3) El Niño Southern Oscillation conditions. The Kilauea Volcano emissions resulted in an aerosol plume that extended westward across the domain driven by easterly trade winds. Aerosol concentrations were anomalously high during both events relative to long-term means and zero-emissions simulations.

The simulated aerosol-cloud interaction signatures common to both events suggested that effects on cloud macro- and microphysical properties with respect to the first indirect effect, such as decreased cloud effective radius from increased CCN, were likely decoupled from local meteorological effects. In contrast, changes in liquid water path and cloud lifetime in the presence of enhanced aerosol concentrations were likely dampened by meteorological variability. Thus even in well-constrained natural experiments as the ones presented in this paper, caution must be taken when drawing conclusions on the second indirect effect without full consideration of meteorological effects.

The 2018 Kilauea degassing event was stronger and more regionally significant with respect to cloud-formation processes. However, when the actual 2008 emissions were increased by 500% (i.e., experiment 2008_5×), computed anomalies in cloud properties were of the same order as for the 2018 event. This and the similarities in the spatial patterns of the anomalies in both events suggests a threshold response of ACIs to overcome meteorological effects, largely controlled by aerosol loading.

For ice clouds, changes in cloud microphysics were significant following the 2018 event while few, if any, effects were apparent in the 2008 event. Our analysis suggested that effects on ice clouds were largely controlled by aerosol injection height and thus the availability of INPs for ice crystal growth and nucleation. Both satellite and model output showed large positive anomalies for ice water path and cloud optical depth during 2018 across the domain, which indicated that conditions favorable for ice cloud formation present in 2018 were absent in 2008. Ash was present in the volcanic plume for both degassing events, but only the plume in 2018 injected volcanic material to sufficiently high altitudes to potentially impact the formation of ice clouds. Sensitivity experiments using increased emissions, plume height, and introducing ash as an ice nucleation particle (INP) suggested that these changes only slightly amplified anomalies for both liquid and ice clouds.

We performed a detailed analysis of the rates of cloud microphysical processes during the two events. As expected, CCN activation played a major role in determining the cloud droplet number concentration for both cases; however, increased liquid droplet concentration did not lead to a substantial reduction in autoconversion rates. Instead, ingestion of CCN within convective parcels may have caused convective invigoration, enhancing the detrainment of condensate in the free troposphere.

As a result, clouds perturbed by aerosols were optically deeper than in the pristine cases. This mechanism also led to enhanced cloud droplet scavenging by accretion.

Microphysical rates in ice clouds were found to be accelerated in 2018 with respect to 2008, likely resulting from higher sea surface temperatures during El Niño conditions in the former (hence more convective inflow to the upper troposphere) and by an increase in convective detrainment in the presence of aerosols. Although the ice nucleation rate in cirrus clouds was substantially higher in 2018 than in 2008, it is likely that it did not control the observed anomaly in ice cloud properties in the former. Accounting for ice formation on ash INP lead to slightly enhanced ice nucleation in warm cirrus clouds thereby facilitating ice sedimentation and decreasing ice water content.

Although the model configuration used in this work presented a clear view of the role of different microphysical processes in determining the evolution of clouds during the 2008 and 2018 events, it precluded a full exploration of the second indirect effect, and the feedbacks between aerosol loading and SSTs. The investigation of these effects requires a coupled ocean-atmosphere model and will be the subject of future work. Chemical effects may also lead to complex interactions between the volcanic plume and the formation of clouds. For example, the ocean entry of 2018 ERZ eruptions caused large clouds of vaporized HCl and water vapor to ascend with the plume (Kern et al., 2020). This was a compositional component absent in the 2008 plume that may lead to the injection of CCN to the upper troposphere as well as enhanced convection in the ERZ region. Accounting for these effects requires a detailed parameterization of the chemical evolution of the volcanic plume, and is left for future studies.

This work showed that satellite observations provided strong evidence for the effects of aerosols modifying clouds during the Kilauea volcanic events. Model simulations elucidated the underlying microphysical processes involved and provided insight into the role of aerosol-cloud interactions in determining the evolution of clouds. We showed that there were many similarities in cloud anomalies during the 2008 and 2018 degassing events and that the discrepancies are largely attributable to differences in aerosol loading. Our work thus provided an unprecedented view of the mechanisms driving the aerosol indirect effect during volcanic events, helping to advance the understanding of the role of aerosol emissions on climate.

*Author contributions.* D. Barahona conceived and supervised this work. K.H. Breen prepared the GEOS simulations and the analysis of the satellite retrievals. H. Bian. and T. Yuan contributed daily emissions files for the volcanic events. S. C. James contributed as a scientific consultant and editor of the manuscript.

*Competing interests.* No competing interests are present.

*Acknowledgements.* D. Barahona was partially supported by the NASA Modeling and Analysis Program, Grant 15-CCST15-0066. Part of this work was performed during K.H. Breen's summer 2019 internship at NASA GSFC. We thank Mian Chin and Kai Yang for providing $SO_2$

emissions data. All authors have read and agreed to the published version of the manuscript. All maps were generated using the NCAR Command Language (Version 6.3.0) Software. (2016). Boulder, Colorado: UCAR/NCAR/CISL/TDD. http://dx.doi.org/10.5065/D6WD3XH5.

## Appendix A:  Ice nucleation on ash particles


Sulfates from $SO_2$ oxidation were the primary aerosol associated with volcanic degassing events. There is, however, evidence that ash was co-emitted with $SO_2$ at the Kilauea Volcano, which may have an effect on the way the plume interacts with clouds by introducing additional surfaces for ice nucleation (ash is typically considered a poor CCN) (Durant et al., 2008). Ash concentration was likely important during the 2018 event when particulate emissions reached the upper troposphere (Neal et al., 2019; Kern et al., 2020). To assess the impact of ash on ice cloud formation, the ice nucleation parameterization in GEOS was amended as follows.


For mixed-phase clouds, ash was assumed to induce immersion freezing. Maters et al. (2019) reported the active site density, $n_s$, of tephra consisting of glassy lithic fragments and ash particles emitted from Kilauea in May 2018. Only ash particles were found to be active ice nucleation sites. For the ash samples, $n_s$ ($cm^{-2}$) was fit as:

$$n_s = \exp\left(-0.6907T + 176.5992\right), \tag{A1}$$

with $240 \leq T \leq 260$ K, where $T$ is temperature. Specific surface area was prescribed as $2.1\,m^2\,g^{-1}$ (Maters et al., 2019). Ash content in the volcanic plume was assumed to be 0.1%, consistent with literature values (Mastin et al., 2009).

Ash was also assumed to heterogeneously nucleate ice in the deposition mode at low temperature ($T < 236$ K) hence affecting the formation of cirrus clouds. Although the ice nucleation efficiency of ash from the Kilauea summit crater has not been reported, Hoyle et al. (2011) measured a saturation freezing threshold of 110% for ash emissions from the Eyjafjallajökull volcano. This value is used as a first approximation for the Kilauea ash emissions. Ash was then treated as a monodisperse INP (Barahona and Nenes, 2009b) and added to the Ullrich et al. (2017) ice nucleation spectrum. The nucleated ice crystal concentration was then calculated using the Barahona and Nenes (2009a) parameterization.

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
