# Peer review of "Effect of volcanic emissions on clouds during the 2008 and 2018 Kilauea degassing events"

_Atmospheric Chemistry and Physics, 2020_

## Referee Comment (RC1) · Anonymous Referee #1 · 13 Nov 2020

This study used satellite observations of aerosol and cloud, and AGCM simulation to investigate aerosol-cloud interaction downwind of the Kilauea volcano during eruptive events of 2008 and 2018. Level-3 MODIS and CALIOP cloud retrievals are used to calculate anomalies in cloud properties vs. climatology during the two events, which are compared with the GEOS model simulations. in the 2018 event with stronger volcanic emissions, significant ($p<0.3$) anomalies in liquid and ice cloud properties are observed from satellites and largely reproduced by the GEOS model, and the simulated anomalies are found to be attributable mostly to the AIE due to volcanic emissions other than meteorological effects. The authors then used the model to illustrate specific pathways how changes in microphysics alter liquid and ice cloud properties.

[Figure]

The topic is within the scope of ACP, the methodology is scientifically sound, and the paper is clearly written and easy to follow. I also found some issues that need further confirmation, especially about how "similar" the two events are. I support the publication of this paper if the following concerns could be addressed.

Major points:

I am a little confused about the current description on changes in cloud properties during the 2008 event, and the "similarity" between the two events. If we look at the summary of MODIS observations in Table 2 and 4, 2 liquid cloud parameters (COD and Reff) show a p-value less than 0.3, whereas all the observed parameters have a p<0.3 during the 2018 event. Simulated changes are also less significant than the simulation for the 2018 events. Interestingly, the authors themselves have conducted the 2008_5x simulation to illustrate that stronger and statistically significant anomalies (similar to the 2018 events) could be derived by enhancing the 2008 emissions, and states that "This and the similarities in spatial patterns for cloud anomalies in JJA 2008 (Figure 6) and MJJ 2018 (Figure 7) suggested a threshold response to overcome meteorological effects."

Therefore, my understanding is that the 2008 event is DIFFERENT from the 2018 event. The observed anomalies are not as strong to overcome possible anomalies from meteorology in liquid cloud properties other than COD and Reff, and the plume cannot reach as high up to modify ice cloud properties. I do agree that this difference is possibly due to the weaker volcanic emissions, which makes the 2018 analysis unique and important. I suggest the authors to modify the description of "similarity" between the two events in the paper (I will also mention several places later), and emphasize their difference and the uniqueness of the 2018 event in the revision.

Minor points:

Page1, Line 8-9: Since the two events are not similar in all aspects, please specify the parameters that are "similarly changed". For example, "Significant changes (p<0.3) in

cloud effective radius and cloud optical depth in both events suggested that. . ."

The presentation flow in the introduction (before Section 1.1) is a little confusing. Paragraph 3 (Page 2, Line 10-20) is detailed description of the AIE, which follows Paragraph 1 more closely. At the same time, Paragraph 2 (Page 2 Line 1-9) introduces AGCM and the difficulty to disentangle meteorology effects when interpreting observations, and I think Paragraph 4 (about the unique conditions Kilauea provides) follows closely from that point. Therefore, I suggest Paragraph 2 and 3 to be swapped.

Page 6, Line 13: In the other simulations where the emission height is changed (2008_PH2km and 2018_PH4km), how are the emissions distributed vertically?

Page 8, Line 10: Add "2008_PH2km" in the sensitivity simulations.

Page 8, Line 10-11: Add "and interpret their differences GEOS simulations."

Page 12, Line 1: This statement only applies to the 2018 event.

Page 12, Line 5: Delete "The plume domain"

Page 12, Line 6: Add "and agrees in magnitudes with the 2018_1x."

Page 17, Line 21: "25.8" disagrees with the number in Table 4 (20.59). Please double check consistency of numbers cited in the text and in the tables.

Page 17, Line 22-23: I am confused. Should this indicate microphysical control is weaker than meteorology for IWP?

Section 5.1, Figures 10-11: Does it make sense to also include the discussion of climatological profiles?

Figure 11: typo in the legend, should be "2018_PH4km"

Page 28, Line 18: suggested to be revised to "the simulated ACI signatures. . ."

Page 29, Line 10: add "and their discrepancies are largely attributable to different magnitudes of volcanic activities and aerosol loadings."

---

## Referee Comment (RC2) · Anonymous Referee #2 · 17 Nov 2020

1- Introduction

This study aims at characterizing the effects of sulfate and ash aerosol emission on macro and microphysical ice and liquid cloud properties during 2 degassing events from the Kilauea volcano in June 2008 and May 2018. The rationale for analysing and comparing these two events is that sulphate aerosols are mainly originated from the volcanic eruption which allows to separate the impacts of aerosol emission on clouds and climate from other confounding effects such as meteorology and anthropogenic emissions. The methodology relies on the comparison between i) anomalies of simulated cloud properties computed against climatology or no volcanic emission experi-

ment and ii) anomalies of cloud properties derived from MODIS and CALIPSO satellites observations, for both 2008 and 2018 events. A cross-comparison of both events is undertaken to characterize the aerosol cloud interaction signatures in response to volcanic aerosol emission and meteorology. The statistic significance of the results is analysed. The main outcomes of this work are

- The increased in cloud condensation nuclei (first indirect effect) in response to increased aerosol concentration was likely decoupled from local meteorology effects while changes in precipitation efficiency and cloud lifetime (second indirect effect) were limited by meteorological variability.

- For the 2008 event aerosol emissions have affected liquid cloud only while the higher injection height of sulfate and ash for the 2018 event leads to modification of ice cloud.
- The injection of volcanic aerosol triggers deepening of the clouds.

- The GEOS model tends to underestimate shallow cloud and overestimate cirrus clouds

- Importance of accounting for ice nucleation on ash particles to represent cirrus processes

This work contributes to a better understanding of the role of meteorology and microphysics in aerosol-cloud interaction processes which represent an important source of uncertainties in current climate models. Overall, the paper is well written and is within the scope of the ACP journal. The methodology and the data are properly presented. The scientific significance of the results is high enough for publication. However, I would suggest addressing the following general remarks and specific comments to improve the quality of the paper prior its publication. Congratulation for this very interesting work.

2- General comments

Could you address the following questions and if needed make the necessary modifications in the text.

- What is the actual contribution of this study to previous works exploiting volcanic emission to characterize the impact of aerosols on ice and liquid cloud processes ? Some indications are given in text, but I suggest to clearly justify the rationales for this study in Introduction and to underline the contributions to previous works in the Abstract and the Conclusion.

- What are the scientific motivations for selecting the Kilauea volcano? I understand that the clean environment allows to separate the impacts of sulfate and ash from other confounding emission sources. Is it the only site offering such "clean" conditions ?

The manuscript suffers from lack of detailed information or clarification, please address my specific remarks (next Section) to improve it.

The authors have chosen to have the results and the discussions in the same section. This is a possible structure for the paper. However in this current form, the result interpretation and discussion are sometimes not enough developed. The back and forth between result description and their interpretation makes the reading difficult (e.g. subsections 5.0.1 and 5.02).

Finally, the limitations of this work need to be acknowledged. Particularly, the uncertainties in the observations used for the analysis have not been discussed or acknowledged properly when comparing satellite retrievals with simulated cloud properties. The discussion does not provide enough insights on the possible shortcomings in the modelling of aerosol-cloud interaction. Recommendations for model improvement could be further developed in the Conclusion.

3- Specific comments

Introduction - Introduction, line 20 page1 : The authors could give some figures illustrating the uncertainties in the radiative impact of ACI (IPCC report). It should be clearly stated in this first paragraph that the understanding of and the representation of ACI in

current models represent a major source of uncertainties for NWP and climate studies (I can see that this is developed in the subsequent paragraph)

- Introduction, page 3: I suggest that the authors provide more rationale on using the Kilauea degassing event compared to other volcanic events. What is the rationale behind ?

- The outline of the paper should be given at the end of Introduction on page 3 this will giove a transition with Section1.

Section 1

- The Section 1 which is dedicated to the volcano description is interesting but may be too long. The authors should better emphasize the differences between the 2008 and 2018 events in terms of injection height, degassing composition and amount, type and duration of eruption...A table could help.

Section 2

- page 5, line 20: It is not clear what processing is applied to missing values, is it gap-filling ?

- the description of the satellite products given in page 5 is not accurate enough.

o What are the variables used in the MODIS and CALIPSO cloud products, is it cloud fraction ? optical depth ? Please list here all the retrieved variables used in your results (a table including all the symbols and acronyms could help).

o What is the vertical resolution of CALIPSO data ?

o What is the temporal frequency of the CALIPSO product ?

o How the anomalies (shown in Fig 5) have been computed for CALIPSO

o What are the rationales for using the MODIS and CALIPSO products ? What are the value-added of each product in term of information content for this work ?

o I would suggest to give some insights on the retrieval algorithm used for each product along with the associated key references (this is partly given for MODIS but missing for CALIPSO)

o what are the uncertainties associated with the MODIS and CALIPSO products ? Could you provide product evaluation references ?

Section 3:

- page 6 line 5 : Which model is it simulating the advection of the aerosol and trace gases: GOCART of GEOS ? What is the type of transport scheme (e.g. semi-lagrangian ?)

- Is GOCART a model embedded in the GEOS model ? How are coupled both models ?

- page 6 line 5-10: A separate paragraph should be dedicated to the aerosol model: type of aerosols, bin size, main simulated processes, key references...This needs to be given before the statements on emission sources.

- page 6, line 9-10: How is volcanic SO2 constrained by OMI data ? (data assimilation ?)

- page 6, line 11-12: could you be more precise on the daily varying emission data set used in this work ? Is it from OMI data as well ? This is not clear

- Is the cloud microphysic scheme a GEOS component ? What is the name of the scheme ?

- Is the GEOS model constrained by data assimilation, particularly for aerosol (MODIS AOD ? . . .)

- the last paragraph (page 6 line 25-35) concerns the model implementation. I suggest having a dedicated subsection 3.2 to model configuration and a subsection 3.1 on general description of the model

[Figure]

Section 4

- page 7 line 22 : which retrieval is used here: cloud fraction, AOD ?

Section 5

- page 9, line 8-12: The following findings are missing from the analysis of Figure 2

o Figure 2 shows a better agreement between the MODIS anomalies and the GEOS anomalies in 2018 compared to 2008. Particularly, the spatial extension of the plume in 2008 is smaller in the simulation than that depicted by the MODIS observations.

o The model anomalies computed against climatology or 0x emission are very similar in 2018 event but not in 2008. Why ?

- page9-10: analysis of Figure 3

o the discrepancies between the simulated anomalies and the MODIS anomalies are larger for cloud fraction than for AOD (Figure 2)

o page 10, line 1: I would de-emphasized "reproduced the spatial distribution": The spatial patterns shown by the simulation are quite different than those shown by MODIS (at least visually, better consistency is shown in the profile).

o page 10, line 5: Why having no correlation between 1x-0x anomaly and retrieval anomalies implies that the observed CF was mainly driven by meteorological variability ? Uncertainties in MODIS observations should also be discussed to put into perspective these findings which strongly rely on the accuracy of the observations.

- Figure 4:

o please could you indicate the meaning of each figure in the caption, we guess that delta SCF and CF refer to the anomalies ?

- Section 5.0.1 :

o This section and the following one are difficult to follow. The back and forth between

results and their interpretation makes the reading quite hard. I would suggest commenting first the results and then interpret them in terms of impact on liquid processes.

o page 12, line 13-15 "suggesting that ACI for 2018 were not limited . . .": why ? additional explanations are needed

o page 12: TWP is used here but defined on page 13. The variables, their symbol, unit and meaning should be defined in the data section and in a table.

- Section 5.0.2: Same remark as for liquid cloud.

Conclusion

- The second paragraph should be improved.. For example, to understand the line 8-10 on page 28 one should know that S02 emission was actually 5 times larger in 2018. One should be able to understand the Conclusion without reading the rest of the text.

4-Technical remarks

- please check the section numbering: a title for section 1 is missing, in section 1 , 1.1.1, 1.1.2 should be replaced by 1.1, 1.2. . .see also section 4

- I suggest to include a Table giving the meaning of the symbols and acronyms.

- There are a lot of acronyms and symbols. I think that for abstract and conclusions the acronyms should be avoided to facilitate the reading.

- Overall the quality of Figures is good.

Please also note the supplement to this comment:
https://acp.copernicus.org/preprints/acp-2020-979/acp-2020-979-RC2-supplement.pdf

---

## Short Comment (SC1) · 10 Dec 2020

Hello,

Most of the details about the SO2 emissions during Kilauea's 2018 eruption should be attributed to Kern et al 2020, rather than Neal et al 2019. Kern et al 2020, which is currently not referenced at all in the paper draft, provides a much more comprehensive analysis of the SO2 emissions during this eruption, as well as estimating some aerosol properties of the gas plume.

Neal et al 2019 provides an extremely general overview of the 2018 eruption, with very

limited (and outdated) estimates of the scope of SO2 degassing.

Here is the Kern et al 2020 citation: Kern, C., Lerner, A. H., Elias, T., Nadeau, P. A., Holland, L., Kelly, P. J., et al. (2020). Quantifying gas emissions associated with the 2018 rift eruption of KÄńlauea Volcano using ground-based DOAS measurements. Bulletin of Volcanology, 82(7), 55. https://doi.org/10.1007/s00445-020-01390-8
* * *

---

## Short Comment (SC2) · 10 Dec 2020

In addition to the role of far greater SO2 and aerosols in the 2018 Kilauea eruption compared to the 2008 eruption, the 2018 eruption involved a very substantial ocean entry (Neal et al 2019) - this is when lava pours into seawater on the coast. During the 2018 eruption, this ocean entry process created large H2O clouds (and also included vaporized HCl and other "laze" plume components) (Kern et al 2020). These water-rich clouds often grew into cumulus rain-bearing cloud systems, that traveled to the WSW. Perhaps the effect of the additional water vaporization and cloud formation during this ocean entry should be better taken into account in the study.

---

## Short Comment (SC3) · 11 Dec 2020

This is an important contribution on the interaction of Kilauea volcanic gas and aerosol emissions with meteorological clouds. In reading through the manuscript, I was left with some questions regarding the SO2 emissions data used in this study. The 2008 and 2018 degassing episodes discussed here differed in two ways that I believe may be important for this discussion. For one, SO2 degassing in 2008 occurred mostly at the summit of the volcano while the majority of degassing in 2018 occurred at lower elevations in the lower East Rift Zone. Also, and perhaps more importantly, the SO2 emission rate during May-July 2018 was approximately an order of magnitude greater

than during 2008. In both cases, I believe the authors have not yet considered the state-of-the-art in our understanding of SO2 degassing to the atmosphere during these eruptive episodes. Below, I've listed a few more details in this regard which I hope might help the authors to further improve their study.

The manuscript (e.g. Figure 1 caption) mentions peak sulfate emissions of 50 kt/d. This is confusing in several ways – for one, we (the USGS) did not measure sulfate, but rather SO2 emissions. High temperature volcanic vents like those at Kilauea emit sulfur mostly in the form of SO2. The SO2 is then converted to sulfate over the course of hours to days. Throughout the manuscript, it is therefore probably best to refer to SO2 emissions rather than sulfate emissions. The 50 kt/d value refers to SO2 and was an estimated minimum value reported by Neal et al. 2019. These emissions occurred mostly from the lower East Rift Zone, not the summit crater shown in the image which is also confusing. Since the publication by Neal et al. in 2019, we have made significant further progress in quantifying the gas emissions related to the 2018 eruption of Kilauea. As Allan Lerner points out in his comment, please refer to Kern et al. 2020 for this information. For example, we now know that peak SO2 emissions of more than 100 kt/d appear to have been sustained throughout the month of June and into early July 2018 (Figure 10 in Kern et al. 2020). We also broadly discuss the topics of aerosol formation and pyrocumulus cloud formation over the lower East Rift Zone, as well as the coincident gas emissions from the volcano's summit and middle East Rift Zone, all of which the authors may find useful in refining thier work.

Regarding the 2008 emissions, please note that Kilauea Volcano was in a state of eruption at its summit Halema'uma'u Crater during the entire 2008-2018 timeframe, not just in 2008. However, the authors are correct in that the highest SO2 emissions (likely > 10 kt/d) occurred during 2008 (see comment below). I would like to encourage the authors to clarify this somewhat, stating that they are focusing on the first year of the 2008-2018 summit eruption during which the highest SO2 emissions occurred, rather than just referring to a 2008 event. I think this would be important given the fact that

emissions averaged about 5 kt/d long after 2008 and continued to have a significant impact on environment and air quality in downwind regions during this entire time. See the following two references on this topic:

Businger S, Huff R, Pattantyus A, Horton KA, Sutton AJ, Elias T, Cherubini T (2015) Observing and Forecasting Vog Dispersion from Kilauea Volcano, Hawaii. Bull Amer Meteor Soc 96:1667–1686. https://doi.org/10.1175/BAMS-D-14-00150.1

Pattantyus AK, Businger S, Howell SG (2018) Review of sulfur dioxide to sulfate aerosol chemistry at Kilauea Volcano, Hawai'i. Atmos Environ 185:262–271. https://doi.org/10.1016/j.atmosenv.2018.04.055

For our best estimates of $SO_2$ emissions during the 2008-2013 period, please refer to our recent data release available here:

Elias, T., Kern, C., Sutton, A.J., and Horton, K., 2020, Sulfur dioxide emission rates from Kilauea Volcano, Hawaii, 2008-2013: U.S. Geological Survey data release, https://doi.org/10.5066/P9K0EZII.

Figure 1A in the above reference provides an overview of $SO_2$ emission rates reported by different authors and using various methods. The estimates vary in magnitude but note that, regardless of the utilized methodology, emissions vastly exceeded the 1,000 t/d level mentioned on page 4, line 17 of the manuscript.

As for the $SO_2$ emissions in 20108, the authors state on page 6, line 10 that they used daily varying $SO_2$ emission rates for their analyses. However, the reference cited is from 2017, so it's unclear where the data corresponding to the 2018 eruption come from. Assuming they come from an analysis of OMI operational $SO_2$ products, it would be quite important to discuss the uncertainty of these data. As described in Kern et al (2020), we had to go to significant effort to account for complex radiative transfer in and around the gas plumes emitted from Kilauea's lower East Rift Zone when analyzing our ground-based DOAS measurements. Similar corrections are likely needed when

analyzing satellite remote sensing observations of these dense gas clouds. As we discuss in the 'Future Work' section of Kern et al 2020, operational satellite products are likely to underestimate the true magnitude of emissions without such corrections. It may therefore be better to use the SO2 emission rates reported in Kern et al 2020 for these analyses (the values are included as a supplement to the paper, along with some measurements of plume height).

Finally, it is also not clear whether it is valid to initialization of the model with the same plume heights for the 2008 and 2018 events, given that the 2008 emissions occurred from the summit of the volcano and the 2018 emissions mostly occurred from the lower East Rift Zone. I would encourage the authors to clarify the assumptions made in their study in this regard, and as one of the reviewers also states, discuss the uncertainties associated with these assumptions in more detail.

Thank you for the opportunity to provide feedback on this effort. I look forward to reading the final version of this important manuscript.

---

## Author Comment (AC1) · 29 Jan 2021

**Response to reviewer comments: Effect of volcanic emissions on clouds during the 2008 and 2018 Kilauea degassing events**

Katherine H. Breen[1, 2], Donifan Barahona[1], Tianle Yuan[1, 3], Huisheng Bian[1, 3], and Scott C. James[4]

[1]NASA, Goddard Space Flight Center, Greenbelt, MD, USA
[2]Universities Space Research Association, Columbia, MD, USA
[3]Joint Center for Earth Systems Technology. University of Maryland, Baltimore County, MD, USA
[4]Baylor University, Departments of Geosciences and Mechanical Engineering, Waco, TX, USA

**Correspondence:** Donifan Barahona (donifan.o.barahona@nasa.gov)

**Review Reports**

The authors thank the reviewers for the thorough comments. Responses are given below.

**Reviewer 1 comments and authors' responses**

5

**Major Points**

**Comment:** *I am a little confused about the current description on changes in cloud properties during the 2008 event, and the "similarity" between the two events. If we look at the summary of MODIS observations in Table 2 and 4, 2 liquid cloud parameters (COD and Reff) show a p-value less than 0.3, whereas all the observed parameters have a $p<0.3$ during the 2018*

10 *event. Simulated changes are also less significant than the simulation for the 2018 events. Interestingly, the authors themselves have conducted the 2008_5x simulation to illustrate that stronger and statistically significant anomalies (similar to the 2018 events) could be derived by enhancing the 2008 emissions, and states that "This and the similarities in spatial patterns for cloud anomalies in JJA 2008 (Figure 6) and MJJ 2018 (Figure 7) suggested a threshold response to overcome meteorological effects"*

15 *Therefore, my understanding is that the 2008 event is DIFFERENT from the 2018 event. The observed anomalies are not as strong to overcome possible anomalies from meteorology in liquid cloud properties other than COD and Reff, and the plume cannot reach as high up to modify ice cloud properties. I do agree that this difference is possibly due to the weaker volcanic emissions, which makes the 2018 analysis unique and important. I suggest the authors to modify the description of "similarity" between the two events in the paper (I will also mention several places later), and emphasize their difference and the uniqueness*

20 *of the 2018 event in the revision.*

**Response:** We aimed to establish whether the difference between the effect of the two volcanic plumes on clouds is solely due to the enhanced amount of material ejected in 2018, or due to differences in the evolution of the plumes. We agree with the reviewer that this point needs clarification. We have added text to describe what is meant by a "similarity in spatial patterns."

In both events the anomalies approximate a Gaussian-like appearance, where the source and maximum zonal mean of the anomaly lies at Kilauea and decreases in strength to the west and north/south, respectively.

We have also added the description for spatial similarity as "anomalies largely constrained by and maximized within the plume domain" to the text. We now explicitly state, with respect to the 2008_5× experiment, that "an increase in aerosol loading of similar magnitude to that observed during MJJ 2018 would have been sufficient to overcome meteorological effects which dampened the 2008 JJA anomalies with respect to long-term behavior."

**Minor Points**

1. **Comment:** *Page1, Line 8-9: Since the two events are not similar in all aspects, please specify the parameters that are "similarly changed". For example, "Significant changes (p<0.3) in cloud effective radius and cloud optical depth in both events suggested that..."*

   **Response:** We have added the suggested text to the abstract.

2. **Comment:** *The presentation flow in the introduction (before Section 1.1) is a little confusing. Paragraph 3 (Page 2, Line 10-20) is detailed description of the AIE, which follows Paragraph 1 more closely. At the same time, Paragraph 2 (Page 2 Line 1-9) introduces AGCM and the difficulty to disentangle meteorology effects when interpreting observations, and I think Paragraph 4 (about the unique conditions Kilauea provides) follows closely from that point. Therefore, I suggest Paragraph 2 and 3 to be swapped.*

   **Response:** Excellent point - we have reorganized the introduction by switching paragraphs 2 and 3, as suggested.

3. **Comment:** *Page 6, Line 13: In the other simulations where the emission height is changed (2008_PH2km and 2018_PH4km), how are the emissions distributed vertically?*

   **Response:** In the default configuration of GEOS, the volcanic plume is injected as a point source at about $1.2$ km, resulting from the assumption that that volcanic emissions height is constrained by the planetary boundary layer ($<$ $2$ km). For the 2008_PH2km and 2018_PH4km runs the base of the plume is assumed still to to be around $1.2$ km but it extends vertically up to $2$ km and $4$ km, respectively. In practice, this means that the volcanic emissions represent a point source at a height about $1/3$ of the top of the plume. The above clarification has been added to the text.

4. **Comment:** *Page 8, Line 10: Add "2008_PH2km" in the sensitivity simulations.*

   **Response:** Done.

5. **Comment:** *Page 8, Line 10-11: Add "and interpret their differences GEOS simulations."*

   **Response:** Done.

6. **Comment:** *Page 12, Line 1: This statement only applies to the 2018 event.*

**Response:** The statement has been modified to " Student's *t*-test statistics indicated that the anomalies were statistically significant, typically to a 70%-80% level ($p < 0.3$, except for AOD and CF in 2008 which are significant only to 60%) for $R_{eff}$ and CDNC more so than for other variables".

7. **Comment:** *Page 12, Line 5: Delete"The plume domain"*

   **Response:** Done.

8. **Comment:** *Page 12, Line 6: Add "and agrees in magnitudes with the 2018_1x."*

   **Response:** Done, and added the following sentence: "The similarity in the magnitude of simulated anomalies for 2008_5× and 2018_1× suggested that increased aerosol loadings would have been sufficient to overcome meteorological effects which dampened the 2008 JJA anomalies with respect to long-term behavior."

9. **Comment:** *Page 17, Line 21: "25.8" disagrees with the number in Table 4 (20.59). Please double check consistency of numbers cited in the text and in the tables.*

   **Response:** Thank you for catching that error. We have checked through the text, and also have made sure that all numbers cited in the text use two decimal places for consistency with the tables.

10. **Comment:** *Page 17, Line 22-23: I am confused. Should this indicate microphysical control is weaker than meteorology for IWP?*

    **Response:** While the GEOS$_{CLIM}$ mimicked the strength of the MODIS anomaly, the zonal mean profile of the GEOS IWP 2018_1×−2018_0× difference had higher correlation with MODIS. We have reworked the text to clarify this.

11. **Comment:** *Section 5.1, Figures 10-11: Does it make sense to also include the discussion of climatological profiles?*

    **Response:** Yes it does. We have included the climatological (excluding 2008 and 2018) profiles in Figures 10 and 11. In general, the climatologies tended to show smaller values of liquid and ice properties than the 0X experiments, with the notable exception of ice mixing ratio and number concentration in 2008. The latter indicating a strong control of the meteorology on the ice phase. This discussion has been added to the text.

12. **Comment:** *Figure 11: typo in the legend, should be "2018_PH4km"*

    **Response:** Corrected.

13. **Comment:** *Page 28, Line 18: suggested to be revised to "the simulated ACI signatures…"*

    **Response:** Done.

14. **Comment:** *Page 29, Line 10: add "and their discrepancies are largely attributable to different magnitudes of volcanic activities and aerosol loadings."*

    **Response:** Done.

---

## Author Comment (AC2) · 29 Jan 2021

**Response to reviewer comments: Effect of volcanic emissions on** clouds during the 2008 and 2018 Kilauea degassing events**

Katherine H. Breen1, 2, Donifan Barahona1, Tianle Yuan1, 3, Huisheng Bian1, 3, and Scott C. James4

1NASA, Goddard Space Flight Center, Greenbelt, MD, USA

2Universities Space Research Association, Columbia, MD, USA

3Joint Center for Earth Systems Technology. University of Maryland, Baltimore County, MD, USA

4Baylor University, Departments of Geosciences and Mechanical Engineering, Waco, TX, USA

Correspondence: Donifan Barahona (donifan.o.barahona@nasa.gov)

**Review Reports**

The authors thank the reviewers for the thorough comments. Responses are given below.

**Reviewer 2 comments and authors' responses**

**5**

**General comments:**

1. **Comment:** What is the actual contribution of this study to previous works exploiting volcanic emission to characterize the impact of aerosols on ice and liquid cloud processes? Some indications are given in text, but I suggest to clearly justify the rationales for this study in Introduction and to underline the contributions to previous works in the Abstract and the Conclusion.

10

**Response:** The following text has been added to clarify our contribution in the context of previous work:

- Abstract:

Although previous studies have assessed the modulation of cloud properties during the 2008 event, this is the first time such an analysis has been reported for the 2018 event and that multiple degassing events have been analyzed and compared at this location.

- Introduction:

The 2008 and 2018 degassing events of Kilauea Volcano have been previously studied with respect to characterization of volcanic and seismic activity during peak emissions (Nadeau et al., 2015; Neal et al., 2019; Elias and Sutton, 2012; Wilson et al., 2008; Orr and Patrick, 2009; Orr et al., 2013; Patrick et al., 2019), analysis of 2008 satellite observational data (Beirle et al., 2014; Mace and Abernathy, 2016; Eguchi et al., 2011; Yuan et al., 2011), comparison of 2008 observed and simulated anomalies (Malavelle et al., 2017), and analysis of the effects of 2018 Kilauea emissions on air quality (Tang et al., 2020).

20

- Conclusion:

Previous analyses have characterized the geologic and atmospheric effects of the 2008 event, but this is the first time that such an analysis has been performed for the 2018 event.

- 2. **Comment:** What are the scientific motivations for selecting the Kilauea volcano? I understand that the clean environment allows to separate the impacts of sulfate and ash from other confounding emission sources. Is it the only site offering such "clean" conditions ?
- **Response:** Not only are local anthropogenic emissions low in the local environment surrounding Kilauea, but because the island is situated on an island approximately 1000 miles from the nearest major anthropogenic emissions source (U.S. West coast), there is a low likelihood of multiple emission sources of similar magnitude. For example, Malavelle et al. (2017) performed a similar analysis of volcanic emissions from a volcano in Holuhraun, Iceland, but could not completely partition volcanic effects from anthropogenic emissions in Western Europe carried by seasonal wind currents. Additionally, Kilauea is well-studied and has had long-term monitoring of emissions, seismic activity, and eruptive behavior over the historical record. Kilauea is a low-altitude volcano, so aerosol injection height is controlled by the strength of the eruption, which provides us with an opportunity to study aerosol-cloud interactions for liquid and ice cloud phases separately. Finally, we clearly take advantage of the unique opportunity of analyzing two events around the same location but with different magnitude in emissions and plume heights. We have clarified this information in the introduction.
- 40 3. **Comment:** The authors have chosen to have the results and the discussions in the same section. This is a possible structure for the paper. However in this current form, the result interpretation and discussion are sometimes not enough developed. The back and forth between result description and their interpretation makes the reading difficult (e.g. subsections 5.0.1 and 5.02).

**Response:** Thank you. We have made an effort to reorganize these sections to increase clarity. Please see detailed responses to comments below.

- 4. **Comment:** Finally, the limitations of this work need to be acknowledged. Particularly, the uncertainties in the observations used for the analysis have not been discussed or acknowledged properly when comparing satellite retrievals with simulated cloud properties. The discussion does not provide enough insights on the possible shortcomings in the modelling of aerosol-cloud interaction. Recommendations for model improvement could be further developed in the Conclusion.
- 50

45

**Response:** We have updated the text to highlight limitations of data products and explicitly cited references. In Section 3 (Data), we point the reader to cited references for uncertainty quantification of satellite retrievals for MODIS variables as

25

30

Uncertainty of MODIS retrievals is discussed in Hubanks et al. (2015) and Bennartz (2007); Bennartz and Rausch (2017) (latter is for CDNC only) ...

, for CALIPSO as

For each level, the gridbox fraction of cloudy, clear, and uncertain areas sum to 1, with the JJA seasonal mean uncertainty  $(2006-2008) \le 0.05$  above the boundary layer (Chepfer et al., 2010) ...

, and OMI

For uncertainty quantification of OMI retrievals, see Carn et al. (2016, 2017); Yang (2017).

Furthermore, partitioning of MODIS cloud phase is discussed in Section 7.1:

Another reason behind the larger aerosol effects on COD and LWP simulated by GEOS than observed by MODIS may lie in differences in phase partitioning (i.e., liquid vs. ice) (Marchant et al., 2016). To help identify thin cirrus clouds, measured top-of-the-atmosphere (TOA) reflectance at 1.38 µm is used to partition high-altitude cirrus clouds from low-altitude liquid clouds. This method is strongly influenced by the relative humidity of the atmospheric column. Therefore, areas with low column water vapor amount may have more clouds partitioned to ice phase than is realistic. This is important because 2008 was a La Niña year and the relative humidity in the atmospheric column was lower than normal, so LWP anomalies may appear low because more clouds were partitioned to the ice phase in the MODIS cloud phase classification algorithm. Even in 2018, most of the total water path (TWP)...

**5. Introduction**

- **Comment:** Introduction, line 20 page 1 : The authors could give some figures illustrating the uncertainties in the radiative impact of ACI (IPCC report). It should be clearly stated in this first paragraph that the understanding of and the representation of ACI in current models represent a major source of uncertainties for NWP and climate studies (I can see that this is developed in the subsequent paragraph)

**Response:** While we agree that it is important to acknowledge GCM uncertainty, we do not feel that another figure would add significant value to this work, given that the suggested figure applies only to a small portion of the introduction and the IPCC report is cited several times within the introduction in reference to GCM uncertainty. However, we have added the following text to the first paragraph of the introduction to summarize topics that are introduced more fully throughout the text:

Additionally, cloud formation is a complex and nuanced physical process occurring on scales far smaller than those resolved by climate models, and the precise feedback mechanisms influencing AIEs on various timescales are not fully understood (Boucher et al., 2013; Klein et al., 2013; Malavelle et al., 2017; Yuan et al., 2011).

55

60

65

70

- **Comment:** Introduction, page 3: I suggest that the authors provide more rationale on using the Kilauea degassing event compared to other volcanic events. What is the rationale behind?

**Response:** Please see the response to the general comment on site selection.

- Comment: The outline of the paper should be given at the end of Introduction on page 3 this will give a transition with Section1.
- 90 **Response:** We have added a brief outline at the end of the introduction as requested.

6. Section 1

- Comment: The Section 1 which is dedicated to the volcano description is interesting but may be too long. The authors should better emphasize the differences between the 2008 and 2018 events in terms of injection height, degassing composition and amount, type and duration of eruption... A table could help.
- 95 **Response:** Thank you for pointing this out. We have condensed the description of events at Kilauea into a single subsection that focuses on the amount of  $SO_2$  emissions, plume height and plume composition for each event, as well as any distinguishing characteristics.

**7. Section 2**

- **Comment:** Page 5, line 20: It is not clear what processing is applied to missing values, is it gap-filling?

Response: Corrected to say: "Missing values in MODIS data, primarily found in CDNC and ice products, were smoothed using cubic spline nearest-neighbor interpolation and a gaussian filter."

- the description of the satellite products given in page 5 is not accurate enough.

- Comment: What are the variables used in the MODIS and CALIPSO cloud products, is it cloud fraction ? optical depth? Please list here all the retrieved variables used in your results (a table including all the symbols and acronyms could help).

Response: Thank you. We defined these paramaters in the Results and Discussion. We have defined the acronyms used in Section 2 (data) and specified which datasets were used from each satellite source. We have also added a lookup table describing each variable.

- Comment: What is the vertical resolution of CALIPSO data ?

**Response:** Corrected to say: "... and vertical resolution typical for most GCMs (40 levels;  $\Delta z = 480$  m)"

- **Comment:** What is the temporal frequency of the CALIPSO product? Response: We used the monthly product. This has been added to the text

- Comment: How the anomalies (shown in Fig 5) have been computed for CALIPSO? Response: They represent the difference between the long term mean (2006-2017, 2008 excluded) and the period of each of the events. This explanation has been added to the text.

115

100

105

110

- Comment: What are the rationales for using the MODIS and CALIPSO products ? What are the value-added of each product in term of information content for this work?

Response: MODIS and CALIPSO are the only instruments that provide retrieved level 3, gridded cloud microphysical properties during the two events, and for which satellite simulators have been developed and implemented. Hence they are essential to validate the modelled cloud properties and the accuracy of the GCM. We have added a paragraph at the beginning of the Section 2 (Data) that introduces the satellite products, which datasets are used, and cites previous works using these products and for which purposes.

- **Comment:** I would suggest to give some insights on the retrieval algorithm used for each product along with the associated key references (this is partly given for MODIS but missing for CALIPSO)

**Response:** We have added the following text to the description of GOCCP data: 125**

Instantaneous profiles of the lidar scattering ratio are computed and used to infer the vertical and horizontal distributions of cloud fraction. For each level, the gridbox fraction of cloudy, clear, and uncertain area sum to 1, with the JJA seasonal mean uncertainty  $(2006-2008) \le 0.05$  above the boundary layer (Chepfer et al., 2010).

- Comment: what are the uncertainties associated with the MODIS and CALIPSO products ? Could you provide product evaluation references ?

**Response:** The references we have already cited provide this information, we have updated the text to clarify.

**8. Section 3**

- Comment: page 6 line 5 : Which model is it simulating the advection of the aerosol and trace gases: GOCART of GEOS? What is the type of transport scheme (e.g. semi-lagrangian?) Is GOCART a model embedded in the GEOS model? How are coupled both models? page 6 line 5-10: A separate paragraph should be dedicated to the aerosol model: type of aerosols, bin size, main simulated processes, key references...This needs to be given before the statements on emission sources.

**Response:** The following paragraph was added to Section 3:

140 Transport of aerosols and gaseous tracers such as CO were simulated using the Goddard Chemistry Aerosol and Radiation model (GOCART) (Colarco et al., 2010), which interactively calculates the transport and evolution of dust, black carbon, organic material, sea salt, and SO2. Dust and sea salt emissions are prognostic whereas  $SO_2$  and biomass burning and antropogenic emissions of SO2, BC, and OC are obtained from the Modern Era Retrospective Reanalysis for Research and Applications-Version 2 (MERRA-2) dataset (Randles et al., 2017). GOCART explicitly calculates the chemical conversion of sulfate precursors (dimethylsulfide or DMS, and SO2) to sulfate. The aging of carbonaceous aerosol is represented by the conversion of hydrophobic to hydrophilic aerosols using a e-folding time of 2 days (Chin et al., 2009). Using the evolving metereological fields from GEOS, each time step GOCART simulates the advection (using a flux-form semi-Lagrangian method, Lin and Rood (1996)), convection, as

120

130

135

well as wet and dry deposition, of aerosol. Aerosol optical depth (AOD) is calculated as a function of aerosol size distribution, refractive indices, and hygroscopic growth. Each aerosol type is assumed to be externally mixed. Size distributions are prescribed for different types, using 5 bins for dust and sea salt, respectively, and single lognormal modes for other aerosol components (Colarco et al., 2010; Chin et al., 2009). This approach was also used to estimate the aerosol number concentration used in the calculation of aerosol-cloud interactions (Barahona et al., 2014).

- Comment: page 6, line 9-10: How is volcanic SO2 constrained by OMI data ? (data assimilation ?)

Response: We have clarified how OMI emissions were used with the following text in Section 3:

Volcanic SO2 emissions are constrained by observations from the Ozone Monitoring Instrument (OMI) on-board NASA's EOS/Aura spacecraft (Carn et al., 2015). For Kilauea, this dataset only provides "constant" annual SO2 emission rates. For the 2008 event, we replaced this data set with daily varying emissions (Carn et al., 2017; Yang, 2017). Daily emissions for 2018 were obtained from Li et al. (2020). Missing values were replaced with Ozone Mapping and Profiling Suite data (https://so2.gsfc.nasa.gov/) whenever possible, otherwise the nearest real data point was used. Vertical column density data were converted from molecules of SO2 cm-2 to kg sulphur per second (kg S s-1) using the linear relationship shown in (Beirle et al., 2014, Fig. 6).

- Comment: page 6, line 11-12: could you be more precise on the daily varying emission data set used in this work ? Is it from OMI data as well ? This is not clear
  - Response: Done.

- Comment: Is the cloud microphysics scheme a GEOS component ? What is the name of the scheme ?

**Response:** Cloud microphysics is part of the "moist" component of GEOS, which includes the evolution of stratiform and convective clouds. The former is described using the Morrison and Gettelman (2008) scheme, sometimes called "MG1". The microphysics of convective clouds are described using the scheme developed in Barahona et al. (2014).

- Comment: Is the GEOS model constrained by data assimilation, particularly for aerosol (MODIS AOD ? ...)

**Response:** Winds and temperature are constrained to the MERRA-2 reanalysis. We run the model in "replay" mode. This technique can be loosely described as a nudging scheme that minimizes the instability and the numerical drifting associated with regular nudging techniques (Takacs et al., 2018). However we didn't constraint water vapor nor aerosol concentrations, even though they are available in MERRA-2. Doing so would have limited the response of clouds to aerosol emissions (via aerosol activation) and vice-versa the response of aerosols to cloud formation and precipitation (via scavenging), adding great difficulty to the analysis. This explanation has been added to the text.

- **Comment:** the last paragraph (page 6 line 25-35) concerns the model implementation. I suggest having a dedicated subsection 3.2 to model configuration and a subsection 3.1 on general description of the model

150

155

165

175

**Response:** We reorganized Section 3 (GEOS model description) as follows. The general model description constitutes the main section content (AGCM description, general description of aerosol transport, microphysical scheme), while a subsection (3.1: Experimental configuration) outlines implementation-specific details (emissions sources/modifications, and constraints).

**Section 4**

- Comment: page 7 line 22 : which retrieval is used here: cloud fraction, AOD ?

**Response:** We obtained level 3 MODIS retrievals of AOD, cloud fraction, effective radius, optical depth, and liquid and ice water path. The Data section has been updated to specify the retrievals used.

**9. Section 5**

- Comment: page 9, line 8-12: The following findings are missing from the analysis of Figure 2
  - **Comment:** Figure 2 shows a better agreement between the MODIS anomalies and the GEOS anomalies in 2018 compared to 2008. Particularly, the spatial extension of the plume in 2008 is smaller in the simulation than that depicted by the MODIS observations.

**Response:** This is an artefact related to the contouring. Had we used different contour ranges for 2008 and 2018 AOD anomalies, this would have highlighted agreement between MODIS and GEOS for each individual event. The zonal anomaly plots (right-most column) show that for each event, MODIS and GEOS spatial distributions are remarkably similar. We have updated the figure using different contouring for the 2008 and the 2018 events.

- **Comment:** The model anomalies computed against climatology or 0x emission are very similar in 2018 event but not in 2008. Why?

**Response:** Essentially because the 2018 anomalies are much larger than in 2008, hence they are much more likely to overcome the natural variability in aerosol emissions, i.e., from passive degassing events and changes in meteorology. We have discussed this point in the paper, and it is now further emphasized in the revised manuscript.

- page9-10: analysis of Figure 3

- **Comment:** The discrepancies between the simulated anomalies and the MODIS anomalies are larger for cloud fraction than for AOD (Figure 2)

**Response:** Yes, this is expected. The AOD anomalies are primarily a function of the aerosol load and largely determined by the volcanic events. Cloud fraction on the other hand is influenced by many factors including convection, sea surface temperature, ENSO state, cloud microphysics and winds, and it is much more sensitive to natural variability. Satellite retrievals are also influenced by empirical definitions of cloudy/non-cloudy regions, adding uncertainty (Pincus et al., 2012). Given this, it is remarkable that the climatological CF is in

185

190

200

195

205

215

relative good agreement with the satellite retrieval demonstrating the skill of GEOS in reproducing clouds during the volcanic events. This point has been further emphasized in the revised manuscript.

- **Comment:** page 10, line 1: I would de-emphasized "reproduced the spatial distribution": The spatial patterns shown by the simulation are quite different than those shown by MODIS (at least visually, better consistency is shown in the profile).

**Response:** We have clarified the text as follows:

... reproduced the spatial distribution of the CF anomaly from the MODIS retrieval (within the plume domain as well as zonal means) ...

- Comment: page 10, line 5: Why having no correlation between 1x-0x anomaly and retrieval anomalies implies that the observed CF was mainly driven by meteorological variability ? Uncertainties in MODIS observations should also be discussed to put into perspective these findings which strongly rely on the accuracy of the observations.

**Response:** Correlation between observations and the 1x-0x anomaly would indicate that volcanic emissions forced ACIs that were decoupled from meteorology - i. e. the observed effects would not have been present without elevated emissions. It is likely that the MODIS CF retrieval has a low bias (although this is in part accounted for by using the satellite simulator) (Pincus et al., 2012), which would move the MODIS anomaly further away from the 1x-0x difference. This is a strong indication that natural variability, instead of the volcanic events, caused the observed anomaly in CF. We have clarified these points in the Section.

- **Comment:** *Figure 4: please could you indicate the meaning of each figure in the caption, we guess that delta SCF and CF refer to the anomalies?*

Response: Done.

- Section 5.0.1:

• **Comment:** This section and the following one are difficult to follow. The back and forth between results and their interpretation makes the reading quite hard. I would suggest commenting first the results and then interpret them in terms of impact on liquid processes.

**Response:** We have made an effort to reorganize this section by describing how results are organized in figures/tables in paragraph 1, presenting the results in paragraphs 2-3, discussion in paragraphs 4-5, and separating the presentation of results/discussion for each event where possible.

• **Comment:** *page 12, line 13-15 "suggesting that ACI for 2018 were not limited ... ": why ? additional explanations are needed*

**Response:** Total water path (TWP) is the sum of liquid water path (LWP) and ice water path (IWP), therefore significant effects in TWP indicate ACIs in both liquid and ice clouds. We have clarified the text as follows: "MODIS TWP anomalies (where TWP is the sum of LWP and IWP) ...".

220

225

230

235

240

• **Comment:** page 12: TWP is used here but defined on page 13. The variables, their symbol, unit and meaning should be defined in the data section and in a table.

**Response:** All datasets/acronyms have been defined in Section 2: Data. We have also added a looku[ table defining each variable.

- Comment: Section 5.0.2: Same remark as for liquid cloud.

**Response:** Thank you for your comment. We have made an effort to rewrite this section to, wherever possible, separate the presentation of results for each event and structure each paragraph such that results are presented first, then discussed.

**Conclusion**

- **Comment:** The second paragraph should be improved.. For example, to understand the line 8-10 on page 28 one should know that S02 emission was actually 5 times larger in 2018. One should be able to understand the Conclusion without reading the rest of the text.

**Response:** Thank you. We have reorganized the conclusions such that overall findings are presented, after which specific findings related to liquid clouds, ice clouds, and microphysical processes are presented separately. We have clarified the text in the example given by the author in the comment above.

**10. Technical Remarks**

Comment: please check the section numbering: a title for section 1 is missing, in section 1, 1.1.1, 1.1.2 should be replaced by 1.1, 1.2... see also section 4

Response: Done.

- Comment: I suggest to include a Table giving the meaning of the symbols and acronyms.

**Response:** We have added a table defining each variable.

 Comment: There are a lot of acronyms and symbols. I think that for abstract and conclusions the acronyms should be avoided to facilitate the reading.

**Response:** No acronyms were used in the abstract. In the conclusions, we have avoided using acronyms except in cases where they were originally used more than once. In such cases, we define the acronym with the first usage.

- Comment: Overall the quality of Figures is good.
- 275 **Response:** Thank you.

260

250

**References**

[revised manuscript text omitted]

---

## Author Comment (AC3) · 29 Jan 2021

**Response to reviewer comments: Effect of volcanic emissions on clouds during the 2008 and 2018 Kilauea degassing events**

Katherine H. Breen1, 2, Donifan Barahona1, Tianle Yuan1, 3, Huisheng Bian1, 3, and Scott C. James4

1NASA, Goddard Space Flight Center, Greenbelt, MD, USA

2Universities Space Research Association, Columbia, MD, USA

3Joint Center for Earth Systems Technology. University of Maryland, Baltimore County, MD, USA

4Baylor University, Departments of Geosciences and Mechanical Engineering, Waco, TX, USA

Correspondence: Donifan Barahona (donifan.o.barahona@nasa.gov)

**Review Reports**

**Short comments and authors' responses**

**Allen Lerner:**

5 We thank Dr. Lerner for taking the time to comment on the manuscript.

- Comment: Most of the details about the SO2 emissions during Kilauea's 2018 eruption should be attributed to Kern et al 2020, rather than Neal et al 2019. Kern et al 2020, which is currently not referenced at all in the paper draft, provides a much more comprehensive analysis of the SO2 emissions during this eruption, as well as estimating some aerosol properties of the gas plume. Neal et al 2019 provides an extremely general overview of the 2018 eruption, with very limited (and outdated) estimates of the scope of SO2 degassing. Here is the Kern et al 2020 citation: Kern, C., Lerner, A. H., Elias, T., Nadeau, P. A., Holland, L., Kelly, P. J., et al. (2020). Quantifying gas emissions associated with the 2018 rift eruption of KÄ 'nlauea Volcano using ground-based DOAS measurements. Bulletin of Volcanology, 82(7), 55. https://doi.org/10.1007/s00445-020-01390-8

**Response:** Thank you. We have added the suggested reference and have included it in our citations, specifically those referring to SO2 emissions.

- Comment: In addition to the role of far greater SO2 and aerosols in the 2018 Kilauea eruption compared to the 2008 eruption, the 2018 eruption involved a very substantial ocean entry (Neal et al 2019) - this is when lava pours into seawater on the coast. During the 2018 eruption, this ocean entry process created large H2O clouds (and also included vaporized HCl and other "laze" plume components) (Kern et al 2020). These water-rich clouds often grew into cumulus rain-bearing cloud systems, that traveled to the WSW. Perhaps the effect of the additional water vaporization and cloud formation during this ocean entry should be better taken into account in the study.

**Response:** This is an excellent point. Unfortunately our current setup is not equipped to simulate this kind of interaction (and as a matter of fact, we know of no model that can explicit simulate such an effect). That being said, our model is

15

20

10

forced using observed sea surface temperatures, and we expect that they would reflect the entry process to some extent, although the horizontal resolution (about 50 km) may be too coarse to elucidate a direct effect on convection, let alone the cloud-chemistry interactions. At present this is a limitation of this work but it would be a very interesting study to conduct in the future. We have included some discussion on this topic in Section 2:

... the ocean entry of 2018 ERZ eruptions caused large clouds of vaporized HCl and water vapor to ascend with the plume (Kern et al., 2020). This was a compositional component absent in the 2008 plume that would increase CCN for liquid clouds in 2018 relative to 2008. We did not include an increase in sea salt aerosols in our parameterization for the 2018 simulations, but recommend this approach for future work.

**References**

35

Kern, C., Lerner, A. H., Elias, T., Nadeau, P. A., Holland, L., Kelly, P. J., Werner, C. A., Clor, L. E., and Cappos, M.: Quantifying gas emissions associated with the 2018 rift eruption of Kilauea Volcano using ground-based DOAS measurements, Bulletin of Volcanology, 82, 1–24, 2020.

3

---

## Author Comment (AC4) · 29 Jan 2021

We have included our response to both comments from Dr. Lerner as a supplement to SC2.

---

## Author Comment (AC5) · 29 Jan 2021

**Response to reviewer comments: Effect of volcanic emissions on clouds during the 2008 and 2018 Kilauea degassing events**

Katherine H. Breen[1, 2], Donifan Barahona[1], Tianle Yuan[1, 3], Huisheng Bian[1, 3], and Scott C. James[4]

[1]NASA, Goddard Space Flight Center, Greenbelt, MD, USA
[2]Universities Space Research Association, Columbia, MD, USA
[3]Joint Center for Earth Systems Technology. University of Maryland, Baltimore County, MD, USA
[4]Baylor University, Departments of Geosciences and Mechanical Engineering, Waco, TX, USA

**Correspondence:** Donifan Barahona (donifan.o.barahona@nasa.gov)

**Review Reports**

**Christoph Kern:**

We thank Dr. Kern for taking the time to comment on the manuscript.

5   **Comment:** *This is an important contribution on the interaction of Kilauea volcanic gas and aerosol emissions with meteorological clouds. In reading through the manuscript, I was left with some questions regarding the SO2 emissions data used in this study. The 2008 and 2018 degassing episodes discussed here differed in two ways that I believe may be important for this discussion. For one, SO2 degassing in 2008 occurred mostly at the summit of the volcano while the majority of degassing in 2018 occurred at lower elevations in the lower East Rift Zone. Also, and perhaps more importantly, the SO2 emission rate*

10   *during May-July 2018 was approximately an order of magnitude greater than during 2008. In both cases, I believe the authors have not yet considered the state-of-the-art in our understanding of SO2 degassing to the atmosphere during these eruptive episodes. Below, I've listed a few more details in this regard which I hope might help the authors to further improve their study.*

**Response:** Thank you. We have added the following text to the "Experimental configuration" subsection to clarify assumptions made about source elevation vs. the importance on injection height in this study:

15        In this work, we assume that the source elevation of emissions (ERZ vs. summit) is irrelevant during peak emissions, but that the injection height of aerosols directly into the troposphere at different altitudes (below or above boundary layer processes) influenced cloud microphysics and macrophysical characteristics for liquid and ice clouds. We also assume the primary aerosol to be $SO_2$ with some percentage of ash, although it is likely that sea salt contributed to ACIs in liquid clouds and is recommended for inclusion in future parameterizations.

20   **Comment:** *The manuscript (e.g. Figure 1 caption) mentions peak sulfate emissions of 50 kt/d. This is confusing in several ways – for one, we (the USGS) did not measure sulfate, but rather SO2 emissions. High temperature volcanic vents like those at Kilauea emit sulfur mostly in the form of SO2. The SO2 is then converted to sulfate over the course of hours to days. Throughout the manuscript, it is therefore probably best to refer to SO2 emissions rather than sulfate emissions. The 50 kt/d value refers to SO2 and was an estimated minimum value reported by Neal et al. 2019. These emissions occurred mostly from*

25 *the lower East Rift Zone, not the summit crater shown in the image which is also confusing. Since the publication by Neal et al. in 2019, we have made significant further progress in quantifying the gas emissions related to the 2018 eruption of Kilauea. As Allan Lerner points out in his comment, please refer to Kern et al. 2020 for this information. For example, we now know that peak SO2 emissions of more than 100 kt/d appear to have been sustained throughout the month of June and into early July 2018 (Figure 10 in Kern et al. 2020). We also broadly discuss the topics of aerosol formation and pyrocumulus cloud formation*
30 *over the lower East Rift Zone, as well as the coincident gas emissions from the volcano's summit and middle East Rift Zone, all of which the authors may find useful in refining their work.*

*Regarding the 2008 emissions, please note that Kilauea Volcano was in a state of eruption at its summit Halema'uma'u Crater during the entire 2008-2018 timeframe, not just in 2008. However, the authors are correct in that the highest SO2 emissions (likely > 10 kt/d) occurred during 2008 (see comment below). I would like to encourage the authors to clarify this*
35 *somewhat, stating that they are focusing on the first year of the 2008-2018 summit eruption during which the highest SO2 emissions occurred, rather than just referring to a 2008 event. I think this would be important given the fact that emissions averaged about 5 kt/d long after 2008 and continued to have a significant impact on environment and air quality in downwind regions during this entire time. See the following two references on this topic:*

40 *Businger S, Huff R, Pattantyus A, Horton KA, Sutton AJ, Elias T, Cherubini T (2015) Observing and Forecasting Vog Dispersion from Kilauea Volcano, Hawaii. Bull Amer Meteor Soc 96:1667–1686. https://doi.org/10.1175/BAMS-D-14-00150.1*

*Pattantyus AK, Businger S, Howell SG (2018) Review of sulfur dioxide to sulfate aerosol chemistry at Kilauea Volcano, Hawai'i. Atmos Environ 185:262–271. https://doi.org/10.1016/j.atmosenv.2018.04.055*
45

*For our best estimates of SO2 emissions during the 2008-2013 period, please refer to our recent data release available here:*

*Elias, T., Kern, C., Sutton, A.J., and Horton, K., 2020, Sulfur dioxide emission rates from Kilauea Volcano, Hawaii, 2008-2013: U.S. Geological Survey data release, https://doi.org/10.5066/P9K0EZII.*
50

**Response:** Thank you for pointing this out. We have clarified the volcanic state of Kilauea during the eruption period like so:

The summit of Kilauea has been in an eruptive state since 2008, and the degassing events of 2008 and 2018 represent brief periods of increased volcanic activity and $SO_2$ emissions resulting in an optically denser plume relative to passive degassing. Several studies have noted the effects of the plume downwind of Kilauea long after
55 violent eruptions have ceased (Businger et al., 2015; Pattantyus et al., 2018), however this study is concerned with ACIs during peak degassing only, when the effects of aerosols on clouds were strongest.

**Comment:** *Figure 1A in the above reference provides an overview of SO2 emission rates reported by different authors and using various methods. The estimates vary in magnitude but note that, regardless of the utilized methodology, emissions vastly exceeded the 1,000 t/d level mentioned on page 4, line 17 of the manuscript.*

*As for the SO2 emissions in 20108, the authors state on page 6, line 10 that they used daily varying SO2 emission rates for their analyses. However, the reference cited is from 2017, so it's unclear where the data corresponding to the 2018 eruption come from. Assuming they come from an analysis of OMI operational SO2 products, it would be quite important to discuss the uncertainty of these data. As described in Kern et al (2020), we had to go to significant effort to account for complex radiative transfer in and around the gas plumes emitted from Kilauea's lower East Rift Zone when analyzing our ground-based DOAS measurements. Similar corrections are likely needed when analyzing satellite remote sensing observations of these dense gas clouds. As we discuss in the 'Future Work' section of Kern et al 2020, operational satellite products are likely to underestimate the true magnitude of emissions without such corrections. It may therefore be better to use the SO2 emission rates reported in Kern et al 2020 for these analyses (the values are included as a supplement to the paper, along with some measurements of plume height).*

**Response:** We have updated the text to clarify the source of 2018 emissions in the Data section like so:

> Daily emissions for 2018 were obtained from Li et al. (2020). Missing values were replaced with Ozone Mapping and Profiling Suite data (https://so2.gsfc.nasa.gov/) whenever possible, otherwise the nearest real data point was used. Vertical column density data were converted from molecules of $SO_2$ $cm^{-2}$ to kg sulphur per second (kg S $s^{-1}$) using the linear relationship shown in (Beirle et al., 2014, Fig. 6).

When addressing this comment, we discovered that May 2018 emissions were left constant during the 2018_1× experiment and therefore repeated the 2018 control simulation using the correct emissions. The correction had only a minor effect on the results and didn't modify our conclusions. This is in part because the effect of aerosol on clouds tends to saturate at high aerosol concentration. Hence even a moderate error in emissions would likely have only a minor impact in our results, particularly for the 2018 event. To maintain consistency with the approach used for the 2008 simulations, we have used OMI data (top down) for 2018 as opposed to the suggested emissions from Kern et al. (2020) (bottom up), which we feel is a more thematic comparison with MODIS observations (also top down).

*Finally, it is also not clear whether it is valid to initialization of the model with the same plume heights for the 2008 and 2018 events, given that the 2008 emissions occurred from the summit of the volcano and the 2018 emissions mostly occurred from the lower East Rift Zone. I would encourage the authors to clarify the assumptions made in their study in this regard, and as one of the reviewers also states, discuss the uncertainties associated with these assumptions in more detail.*

*Thank you for the opportunity to provide feedback on this effort. I look forward to reading the final version of this important manuscript.*

**Response:** Thank you for your response - the feedback has provided us with an opportunity to address a few important assumptions. On the topic of plume height, here we are interested in to top of the plume because it is the maximum altitude at which aerosols are directly injected into the atmospheric column. For the sensitivity experiments, we tested if direct interaction

with aerosols above vs. below the boundary layer had significant effects on cloud processes, specifically ice clouds. In 2018, the simulated and observed ACIs showed more similarities than ice clouds in 2008 - in fact, 2008 ice clouds show little to no anomalous cloud processes. This was an interesting finding during the initial research phase and prompted us to ask - why? We knew that emissions were higher in 2018 than 2008, but it had also been reported that plumes had been observed as high as 8 km during peak 2018 emissions. Intuitively, it makes sense that injecting SO2 and ash directly at high latitudes would increase the concentration of ice nucleating particles originating from ash. From our sensitivity experiments, it appears that this intuitive assumption was, at least in part, correct. We recommend the parameterization of other possible "culprits" for future work.

**References**

Beirle, S., Hörmann, C., Penning de Vries, M., Dörner, S., Kern, C., and Wagner, T.: Estimating the volcanic emission rate and atmospheric
lifetime of $SO_2$ from space: a case study for Kīlauea Volcano, Hawai 'i, Atmospheric Chemistry and Physics, 14, 8309–8322, 2014.

Businger, S., Huff, R., Pattantyus, A., Horton, K., Sutton, A. J., Elias, T., and Cherubini, T.: Observing and forecasting Vog dispersion from
Kīlauea Volcano, Hawaii, Bulletin of the American Meteorological Society, 96, 1667–1686, 2015.

Kern, C., Lerner, A. H., Elias, T., Nadeau, P. A., Holland, L., Kelly, P. J., Werner, C. A., Clor, L. E., and Cappos, M.: Quantifying gas
emissions associated with the 2018 rift eruption of Kīlauea Volcano using ground-based DOAS measurements, Bulletin of Volcanology,
82, 1–24, 2020.

Li, C., Krotkov, N. A., and Leonard, P.: OMI/Aura Sulfur Dioxide ($SO_2$) Total Column L3 1 day Best Pixel in 0.25 degree × 0.25 degree V3,
https://doi.org/10.5067/Aura/OMI/DATA3008, 2020.

Pattantyus, A. K., Businger, S., and Howell, S. G.: Review of sulfur dioxide to sulfate aerosol chemistry at Kīlauea Volcano, Hawai 'i,
Atmospheric Environment, 185, 262–271, 2018.

---

## Author Response (AR2)

**Response to reviewer comments: Effect of volcanic emissions on clouds during the 2008 and 2018 Kilauea degassing events**

Katherine H. Breen[1, 2], Donifan Barahona[1], Tianle Yuan[1, 3], Huisheng Bian[1, 3], and Scott C. James[4]

[1]NASA, Goddard Space Flight Center, Greenbelt, MD, USA
[2]Universities Space Research Association, Columbia, MD, USA
[3]Joint Center for Earth Systems Technology. University of Maryland, Baltimore County, MD, USA
[4]Baylor University, Departments of Geosciences and Mechanical Engineering, Waco, TX, USA

**Correspondence:** Donifan Barahona (donifan.o.barahona@nasa.gov)

**Review Reports**

The authors thank the reviewers for the thorough comments. Responses are given below.

**Reviewer 2 comments and authors' responses**

5

*Congratulation for this work. the revision has substantially improved the quality of the paper. Most of my comments have been properly addressed and I recommend the paper for publication. However, I would like the authors to clarify the following points:*

**Comment:** *regarding the use of satellite observation in the model: I understand that OMI data is used to feed the model with*
10 *volcanic S02 emission. But the answer to my question on the use of data assimilation of satellite observation to estimate aerosol concentrations or other state variables is not clear. I suggest to clearly state this in the model section. Besides, one needs to distinguish between data assimilation used to constrain the time trajectory of the model and the use of satellite observation as a forcing variable (emission)*

**Response:** Simulation of aerosols is described in Section 3.1 as follows:

15      Transport of aerosols and gaseous tracers such as CO were simulated using the Goddard Chemistry Aerosol and
        Radiation model (GOCART) (Colarco et. al., 2010), which interactively calculates the transport and evolution
        of dust, black carbon, organic material, sea salt, and $SO_2$. Dust and sea salt emissions are prognostic whereas
        biomass burning and antropogenic emissions of $SO_2$, black carbon, and organic carbon are obtained from the
        Modern Era Retrospective Reanalysis for Research and Applications-Version 2 (MERRA-2) dataset (Randles et.
20      al, 2017). . . . Using the evolving meteorological fields from GEOS, for each time step GOCART simulates the
        advection (using a flux-form semi-Lagrangian method, (Lin et. al., 1996)), convective transport, and the wet and
        dry deposition of aerosol tracers.

The nudging of the model state using the "replay" technique is explained in section 3.2 as follows:

To account for model drift all simulations were run in "replay" mode, where pre-computed analysis increments from MERRA-2 were applied to nudge the model state (i.e., horizontal winds and temperature) to the reanalysis every six hours.

To address the reviewer's concern, we have expanded the description in Section 3.2 to clarify the difference between nudging the model state, which indirectly affects the aerosol, and directly nudging aerosol concentrations:

. . . **Aerosol concentrations are indirectly constrained by the reanalysis since their transport and evolution, as well as the emission of dust and sea salt, depend on the model state (i.e., winds and temperature). The emission of sulfate precursors (SO2) is constrained using satellite retrievals as described in Section 3.1. However aerosol concentrations were not directly nudged to the MERRA-2 product, even though the aerosol increments are also available (?). Doing so would have limited the response of clouds to aerosol (via aerosol activation) and vice-versa, the response of aerosols to cloud formation and precipitation (via scavenging).**

**Comment:** *The authors have not properly addressed my first comment about "What is the actual contribution of this study to previous works exploiting volcanic emission to characterize the impact of aerosols on ice and liquid cloud processes" . The justification wrt to Kilauea event is fine, but the readers expect a clear understanding of what is the contribution of the paper to the more general topic on the interactions between volcanic emission and aerosol-cloud interaction processes.*

**Response:** We have clarified the contribution of this work in the Introduction as follows:

. . . However analyses of ice clouds and of the impacts of the 2018 event on cloud properties and evolution has not yet been reported. This work constitutes a substantial contribution towards identifying ACI signatures for liquid and ice macro and microphysical processes and their sensitivity to aerosol loadings and CCN/INPs.

**Comment:** *the following statement added for CALIPSO is not clear : For each level, the gridbox fraction of cloudy, clear, and uncertain areas sum to 1, with the JJA seasonal mean uncertainty (2006–2008) ≤ 0.05 above the boundary layer (Chepfer et al., 2010) . . .*

**Response:** We have clarified the statement about GOCCP cloud fraction uncertainty as follows:

. . . The seasonal mean uncertainty (2006–2008) for GOCCP cloud fraction is $\leq 0.05$ above the boundary layer (Chepfer et al., 2010).